# Basal control of supraglacial meltwater catchments on the Greenland Ice Sheet

Josh Crozier[1], Leif Karlstrom[1], and Kang Yang[2,3]

[1]University of Oregon Department of Earth Sciences, Eugene, Oregon, USA
[2]School of Geography and Ocean Science, Nanjing University, Nanjing 210023, China
[3]Joint Center for Global Change Studies, Beijing 100875, China

**Correspondence:** Josh Crozier (crozierjosh1@gmail.com)

**Abstract.** Ice surface topography controls the routing of surface meltwater generated in the ablation zones of glaciers and ice sheets. Meltwater routing is a direct source of ice mass loss, as well as a primary influence on subglacial hydrology and basal sliding of the ice sheet. Although the processes that determine ice sheet topography at the largest scales are known, controls on the topographic features that influence meltwater routing at supraglacial internally-drained-catchment (IDC) scales (< 10s of km) are less well constrained. Here we examine the effects of two processes on ice sheet surface topography: transfer of bed topography to the surface of flowing ice and thermal-fluvial erosion by supraglacial meltwater streams. We implement 2D basal transfer functions in seven study regions of the western Greenland Ice Sheet ablation zone using recent data sets for bed elevation, ice surface elevation, and ice surface velocities. We find that ∼1-10 km scale ice surface features can be well-explained by bed topography transfer in regions with different multi-year averaged ice flow conditions. We use flow-routing algorithms to extract supraglacial stream networks from 2-5 m resolution digital elevation models, and compare these with synthetic flow networks calculated on ice surfaces predicted by bed topography transfer. Multiple geomorphological metrics calculated for these networks suggest that bed topography can explain general ∼1-10 km scale supraglacial meltwater routing, and that thermal-fluvial erosion thus has a lesser role in shaping ice surface topography on these scales. We then use bed topography transfer functions and flow-routing to conduct a parameter study predicting how supraglacial internally drained catchment (IDC) configurations and subglacial hydraulic potential would change under varying multi-year averaged ice flow and basal sliding regimes. Predicted changes to subglacial hydraulic flow pathways directly caused by changing ice surface topography are subtle, but temporal changes in basal sliding or ice thickness have potentially significant influences on IDC spatial distribution. We suggest that changes to IDC size and number density could affect subglacial hydrology primarily by dispersing the englacial/subglacial input of surface meltwater.

# 1 Introduction

During warmer months on the Greenland ice sheet, surface melting in the ablation zone generates a large volume of water. Some meltwater is stored in or flows through porous firn or weathered ice, but most flows across the ice surface forming networks of supraglacial streams and lakes (such as the stream network shown in Fig. 1.B) (Fountain and Walder, 1998; van den Broeke et al., 2009; Andersen et al., 2015). The majority of these streams feed into the englacial and subglacial hydrological systems either by flowing directly into open moulins (e.g., Chu, 2014; Smith et al., 2015), or by flowing into supraglacial lakes which can drain when enough water pressure builds up to hydraulically fracture the ice (Das et al., 2008; Selmes et al., 2011; Stevens et al., 2015). Much of the meltwater will ultimately end up in the ocean (Enderlin et al., 2014; Andersen et al., 2015). Along the way, subglacial water and temporal variations in subglacial water flux can significantly influence ice advection by modulating basal sliding resistance (e.g., Zwally et al., 2002; Schoof, 2010; Sole et al., 2011; Shannon et al., 2013; Tedstone et al., 2014). The spatial and temporal flux of surface meltwater to the subglacial hydrological system, how this flux evolves with changing climate and/or ice flow, and how subglacial hydraulic pathways evolve in response to meltwater input, are all poorly constrained and largely not incorporated into current ice sheet mass balance models (Larour et al., 2012; Gillet-Chaulet et al., 2012; Gagliardini et al., 2013; Lipscomb et al., 2013; Khan et al., 2015; Smith et al., 2017).

Ice sheet surface meltwater flows downhill as dictated by surface topography. The largest scale of Greenland Ice Sheet topography is a continental-scale ($\sim 1000$ km) gravity current profile, where average surface slopes are very gradual in the interior of the ice sheet (on the order of $10^{-2.5}$ radians or less) and steepen approaching the margins (on the order of $10^{-2}$ radians in our ablation zone study regions) (Cuffey and Paterson, 2010). Deviations from this geometry at smaller wavelengths, however, reflect a combination of other physical processes. Some are products of the surface energy balance, such as solar radiation-driven ice melting/sublimation, melting of ice by flowing surface water (we will refer to this process as thermal-fluvial incision), and snow accumulation (Cuffey and Paterson, 2010; Karlstrom and Yang, 2016; Boisvert et al., 2017; Meyer and Hewitt, 2017). Others are products of ice flow processes such as crevassing (Echelmeyer et al., 1991; Cuffey and Paterson, 2010), propagating ice flux waves (Weertman, 1958; Nye, 1960; van de Wal and Oerlemans, 1995; Hewitt and Fowler, 2008), and the transfer of spatially variable bed topography, basal sliding, and ice rheology (due to temperature, grain alignment, or impurities) to the surface (Gudmundsson, 2003; Raymond and Gudmundsson, 2009; Sergienko, 2013; Graham et al., 2017).

The advection of ice over rough bed topography (such as that shown in Fig. 1.A) (Budd, 1970; Hutter et al., 1981; Gudmundsson, 2003; De Rydt et al., 2013; Joughin et al., 2013) is thought to be a significant source of IDC scale ($\sim$1-10 km) ice surface topography, and is a primary focus of our study. Supraglacial IDC and lake locations generally remain fixed year to year despite ice advection, which suggests a basal controlling process (Lampkin, 2011; Lampkin and van der Berg, 2011; Selmes et al., 2011; Sergienko, 2013; Ádám Ignéczi et al., 2016; Karlstrom and Yang, 2016). We use the term "bed" loosely to refer to whatever material composes the substrate under an ice sheet. In many locations the bed contains a deformable till layer which may not influence ice flow in the same way as rigid bedrock (Tulaczyk et al., 2000; Cuffey and Paterson, 2010), and bedrock erodes under the action of ice motion (Sugden, 1978; Hart, 1995).

Thermal-fluvial incision is also important for the evolution of surface topography and meltwater channel networks (e.g., Parker, 1975). Surface melt rates in many areas of the Greenland Ice Sheet ablation zone are greater than 1 m/yr (Noel et al., 2015); stream channels can be meters deep, and are in places observed to flow in directions not parallel to the surrounding ice surface slope or to slice through topographic ridges (Smith et al., 2015; Yang et al., 2015). Karlstrom and Yang (2016) suggested

that longitudinal elevation profiles of supraglacial streams might even be inverted for primary production rate of meltwater, the analog to inferring climate variations and tectonic uplift rates from river profiles in terrestrial settings. However, although thermal-fluvial incision is required to make channels in the first place (e.g., lowering rate in channels must be greater than surroundings), it is unclear whether dynamic stream incision is efficient enough compared to other topographic influences to significantly affect IDC-scale topography and meltwater routing (Karlstrom and Yang, 2016). In this way supraglacial streams

may be more analogous to ephemeral gullies on earth flows (Mackey and Roering, 2011) than to terrestrial river networks. The ice surface in ablation zones advects stream channels horizontally at velocities greater than 100 m/yr (Joughin et al., 2010b, a; Nagler et al., 2015), deforming or offsetting stream networks as they incise. This has been observed where Greenland supraglacial stream channels form along offset but parallel pathways as channels from previous years are advected out of topographic lows during winter months, though there are also stream channels that are reused for multiple years (Karlstrom

and Yang, 2016).

Understanding the relative contributions of processes that govern ablation zone surface topography should yield better predictions of meltwater routing through time. Here, we use multiple data sets to examine the effects and significance of bed topography transfer and thermal-fluvial incision on ice sheet surface topography and meltwater routing. Ice surface velocity measurements, high resolution ice surface imagery and digital elevation models (DEMs), and bed elevation DEMs are now

concurrently available over large expanses of the Greenland ice sheet ablation zone (Joughin et al., 2010b, a; Helm et al., 2014; Morlighem et al., 2017a, b; Nagler et al., 2015; Noel et al., 2015; ArcticDEM, 2017). Many of these data sets are rapidly increasing in quality and temporal coverage, and developing methods to efficiently integrate such large data sets is thus important.

We implement approximate analytical solutions for bed topography transfer through flowing ice (Gudmundsson, 2003) over

2D regions of the Greenland ablation zone, evaluating the extent to which this transfer can explain observed ice surface topography as a function of wavelength. To examine what influences supraglacial meltwater routing, we apply flow-routing algorithms both to ice surface DEMs and to synthetic ice surfaces predicted from modeling bed topography transfer. In the resulting flow networks we examine channel slope versus accumulated drainage area trends to assess the fluvial erosion signature, and we examine steam network conformity with surrounding ice surface topography to quantify the importance of different wavelengths

for explaining stream network spatial structure. We identify bed topography transfer as the primary control on IDC-scale (~1-10 km) surface topography and meltwater routing, and then use bed topography transfer functions to predict how Greenland surface IDC configuration and subglacial hydraulic flow pathways would change in response to varying ice flow conditions.

## 2 Methods

### 2.1 Data

We use stereo imagery derived ArcticDEM 2-5 m resolution mosaics for 2011 Greenland Ice Sheet surface elevation (Arctic-DEM, 2017; Noh and Howat, 2015). These DEMs were created by piecing together smaller DEM strips that in some cases come from data taken over multiple months. This is a potential source of error in our analysis since ice sheet surface topography can vary temporally due to a variety of processes including horizontal ice advection (on the order of 100 m/yr in our study areas (Joughin et al., 2010b, a; Nagler et al., 2015)), ablation (on the order of 1 m/yr (Bartholomew et al., 2011)), accumulation (on the order of 1 m/yr (Koenig et al., 2016)), and advection-related thickening/thinning such as that caused by changes in basal properties (on the order of 1 m/yr (Das et al., 2008; Helm et al., 2014)). In our study regions we observe $< \sim 1$ m vertical and $< \sim 10$ m horizontal offsets from surface DEM stitching (where different raw source data sets are combined).

We use the Icebridge BedMachine v3 150 m resolution Greenland bed elevation DEM (Morlighem et al., 2017a, b). This product is derived from radar data, and in some regions also from ice mass conservation modeling. This product has large error (as much as 500 m) in areas with low radar pass density; we selected study regions with a range of bed DEM quality (shown in Fig. 2). Many regions of the ablation zone, including our study regions, are where mass conservation modeling was used to extrapolate raw radar transects into contiguous bedrock DEMs. This approach and its advantages are explained in detail by Morlighem et al. (2011) and Morlighem et al. (2014). Surface elevations, surface velocities, and mass balances estimates are used to produce more accurate bed DEMs that are consistent with multiple radar-derived data sets which have limited spatial coverage (as shown in Fig. 2). The mass conservation modeling does not preclude us from using these DEMs to evaluate the effectiveness of bed topography transfer functions at predicting surface topography, since the approach used in creating the bed DEMs only solves mass conservation equations and does not take into account the momentum balance accounted for by the transfer functions (described in Section 2.3.1). However, as an additional precaution, we focus our analysis primarily on regions with more dense radar transect coverage. DEMs in these regions should most closely reflect the raw radar data, and also generally have higher effective resolution.

We use 2009 InSAR derived MEaSUREs (Joughin et al., 2010b, a) for 500 m resolution Greenland winter ice surface velocities. We use Landsat imagery to identify moulins, lakes, and stream channels (Yang and Smith, 2016). We use RACMO 2.3p2 at 1 km resolution for melt data from the full year 2015 (Noel et al., 2015) to indicate relative melting between different regions of Greenland. All data sets do not necessarily correspond temporally, which is a potential source of error in our analysis since ice velocity, ice surface topography, and bed topography can vary temporally (Sugden, 1978; Hart, 1995; Bartholomew et al., 2011; Sole et al., 2011; Helm et al., 2014). We focus our analysis and discussions on multi-year averaged ice flow properties, and do not attempt to model seasonal dynamics.

### 2.2 Study regions

We focus on areas of the Greenland Ice Sheet ablation zone that exhibit significant supraglacial drainage networks, are not heavily crevassed, and do not contain ice streams (pathways where ice is advecting very rapidly relative to the surrounding

regions of an ice sheet). We additionally require areas with high resolution (2-5 m) surface DEMs and near-uniform ice surface velocities. Using these criteria we select seven internally drained catchments (IDCs) from the western Greenland Ice Sheet as primary study regions (shown in Fig. 1.A, with additional information in Table 1). These regions cover a significant range of the elevations, ice thicknesses, ice surface slopes, and ice surface velocities over which extensive supraglacial stream networks

form on western Greenland.

## 2.3  Bed topography and basal sliding transfer

### 2.3.1  Linear transfer functions

The governing equations of flowing ice are the Stokes equations for conservation of momentum and mass balance for an incompressible fluid. Ice is often described with a nonlinear constitutive relation know as Glen's law (Cuffey and Paterson, 2010), but

here a linear Newtonian ice rheology is assumed. Ice rheology is also assumed to be spatially and temporally constant, though the rheology of ice generally varies with temperature, grain geometry, and impurities. The momentum equations we solve are

$$\boldsymbol{\nabla} p = \eta \boldsymbol{\nabla}^2 \boldsymbol{u} + \rho_i \boldsymbol{g}, \tag{1}$$

where $p$ is pressure, $\eta$ is effective dynamic ice viscosity, $\boldsymbol{u}$ is ice velocity, and $\rho_i$ is ice density. Cartesian coordinates are used, where the $z$-axis is aligned normal to the mean ice surface and the $x$-axis points in the direction of maximum bed gradient. $\boldsymbol{g}$

is the gravitational acceleration (in the $-z$ direction). Conservation of mass is given by

$$\boldsymbol{\nabla} \cdot \boldsymbol{u} = 0. \tag{2}$$

Linear stability analysis of Eqs. (1) and (2) by Gudmundsson (2003) provides analytical transfer functions that predict approximate ice surface topography over underlying rough bed topography or basal sliding variations. In the spectral domain, transfer functions are generally of the form $\hat{X}_o(\boldsymbol{k}) = \hat{X}_i(\boldsymbol{k})\hat{T}(\boldsymbol{k})$ where $\boldsymbol{k} = (k_x, k_y)$ is a wavenumber (inverse wavelength)

vector, $X_o$ is output data (ice surface elevation in our case), $X_i$ is input data (bed elevation in our case), and $T$ is the transfer function relating outputs to inputs. Transfer functions are possible to obtain for linear time-invariant systems; the basic underlying principals are that the output for such systems may be calculated for any single wavenumber input, that any input may be represented as a sum of individual wavenumber components via Fourier transform, and that the output will be a sum of the independent outputs from each input component (Stein and Wysession, 2005). In the rest of this section we will summarize the

derivation of the transfer functions (described fully in Gudmundsson (2003)) and the important approximations made in this derivation.

Bed elevation is assumed to not change temporally beyond an initial perturbation. Basal melting/freezing are also ignored, assumptions that are likely reasonable from a mass conservation perspective due to the generally slow rates of basal melting/freezing (Huybrechts, 1996). Basal sliding velocity $\boldsymbol{u}_b$ is assumed to be governed by a sliding law of the form

$$\boldsymbol{u}_b(x,y) = C(x,y)\boldsymbol{\tau}_b(x,y) \tag{3}$$

where $\boldsymbol{\tau}_b$ is basal shear stress and $C(x,y)$ is a sliding parameter. We will often refer to non-dimensionalized basal sliding coefficient $C^*(x,y) = C(x,y)\frac{2\eta}{H}$ (approximately equivalent to slip ratio, the ratio of basal sliding velocity to ice deformational velocity). Other forms of sliding law have been proposed (Fowler, 1986; Tulaczyk et al., 2000; Cuffey and Paterson, 2010). The basal boundary condition (at the bed-ice interface) combines this sliding law with a no-flow condition dictating zero ice

velocity normal to the boundary. Surface accumulation/ablation are ignored, which is reasonable as both rates are generally small compared to ice advection rates (van den Broeke et al., 2011). The ice surface boundary conditions are zero traction plus the kinematic boundary condition

$$\frac{\partial Z}{\partial t} = u_z - u_x \frac{\partial Z}{\partial x} - u_y \frac{\partial Z}{\partial y}. \tag{4}$$

Thus the ice surface boundary is the only source of time variation in the system.

Parameters including ice thickness, surface velocity, and surface slope are assumed to be similar over the domain of interest, which allows for solutions to be obtained as perturbations to a zeroth-order infinite plane slab solution. In order for these assumptions to be valid, it is assumed that bed topography amplitude is much smaller than ice thickness, and that the domain of interest is small compared to the horizontal dimensions of the ice sheet. The zeroth-order ice surface $Z^0$ is a plane with slope $\alpha$ in the direction of ice flow. Zeroth-order ice thickness $H$ is the mean ice thickness in the domain. Zeroth-order basal shear

stress is given by $\tau_b = \rho g H \sin(\alpha)$, and zeroth-order deformational velocity is given by $U_d = \frac{1}{2\eta}\tau_b H$, where $\rho$ is (spatially constant) ice density.

Bed elevation $B$ is expressed as $B = B^0 + \epsilon F_B^\epsilon$, where $B^0$ is zeroth-order (horizontal plane) bed elevation, $F_B^\epsilon$ represents perturbations to $B^0$, and $\epsilon = $ bed topography amplitude$/H \ll 1$. Basal sliding coefficient $C$ is similarly expressed as $C = C^0 + \beta F_C^\beta$ where $0 \leq \beta \ll 1$. Equations 1 and 2 are linearized around $\epsilon, \beta = 0$ and solved in the Fourier domain. Ice surface

elevation is then given by

$$Z = Z^0 + \epsilon F_Z^\epsilon + \beta F_Z^\beta + \mathcal{O}(\epsilon^2, \beta^2, \epsilon\beta), \tag{5}$$

where $\epsilon F_Z^\epsilon$ and $\beta F_Z^\beta$ represent the first order (linear) ice surface response to $B$ and $C$ perturbations. Higher order terms $\mathcal{O}(\epsilon^2, \beta^2, \epsilon\beta)$ are discarded.

The full time-dependent transfer functions can be found in Gudmundsson (2003). A steady state surface configuration to

bed topography and basal sliding perturbations is approached as $t \to \infty$. We note that ice flow parameters in the transfer functions are not strictly independent, such that there are restricted parameter combinations that correspond to real ice flow configurations.

Although the linear transfer functions derived by Gudmundsson (2003) do not capture all the complexities of ice motion, they have some significant advantages over other methods for solving our desired ice flow problem. They do not make a shallow

ice approximation (which ignores longitudinal stresses and thus breaks down at length scales on the order of ice thickness $H$ (e.g., Cuffey and Paterson, 2010)), and are thus valid at spatial scales $< H$. Additionally, they can be efficiently implemented over 2D IDC-scale regions without requiring initial conditions, flow line geometry, and domain-edge boundary conditions that many numerical flow simulators need. We will show that the functions reproduce general topographic features and amplitude spectra of our Greenland Ice Sheet study regions well, and thus provide a useful predictive tool.

### 2.3.2 Transfer function implementation

When implementing the transfer functions in all following analysis we will assume the ice surface has reached a steady state in response to the underlying bed topography and basal sliding conditions. To examine the validity this assumption, we calculate the transfer function perturbation adjustment timescales for parameters representative of the western Greenland ablation zone (from study region R1 (Fig. 1.A, Table 1) with $C^{0*} = 10$ and $\eta = 10^{14}$ Pa s) using the time-dependent transfer functions defined in Gudmundsson (2003). For the range of ice flow parameters we are interested in, there is no appreciable downstream advection of surface perturbations, and so the surface response soon after a basal perturbation is essentially a lower amplitude scaling of the steady state (maximum amplitude) surface response. We find that the time scale for bed topography or basal sliding transfer amplitudes to reach 95% of their steady state values is as much as 60 years for the longest wavelengths of topography in our typical study areas ($\sim$20 km), and is $\sim$3-20 years for wavelengths that typically exhibit the highest transfer ($\sim$1-10 km). It is unlikely that bed topography, ice sheet thickness, or ice sheet surface slope change significantly over these timescales, but ice velocity and basal sliding can vary on day to year timescales, meaning that the steady state assumption is a potential source of error in our analysis (Das et al., 2008; Bartholomew et al., 2011; Sole et al., 2011; Helm et al., 2014; Chandler et al., 2013; Tedstone et al., 2014).

Methods have recently been developed and applied for implementing the linear basal transfer functions along flowlines with spatially varying parameters (Igneczi et al., 2018; Ng et al., 2018), which allows for implementation of the transfer functions over large regions. However, implementing the transfer functions just along flowlines can result in significant inaccuracy. With ice flow parameters representative of the western Greenland Ice Sheet ablation zone, the transfer amplitude of IDC-scale ($\sim$1-10 km) bed features predicted by the linear transfer functions could vary by up to a factor of 10 depending upon the 2D alignment of those features (see supplement). Our approach retains the simpler constant-parameter model but accounts for 2D effects. We implement the basal transfer functions over rectangular domains of small enough size ( 20 km across) that ice flow parameters are relatively uniform within each domain.

To implement the transfer functions in (east, north, vertical) Cartesian coordinates, we calculate absolute wavenumbers as $k = \|\boldsymbol{k}\|$ and wavenumbers in the ice flow direction as $k_U = \frac{\boldsymbol{k} \bullet \boldsymbol{U}}{\|\boldsymbol{U}\|}$. We can then calculate transfer function matrices $\hat{T}_B(k_x, k_y)$ and $\hat{T}_C(k_x, k_y)$ corresponding to the discrete wavenumber components of a given bed DEM. The amplitude matrices $\left(|\hat{T}_{B,C}|\right)$ are symmetric about the line perpendicular to the ice flow direction, and the phase matrices $\left(\arg(\hat{T}_{B,C})\right)$ are anti-symmetric about this line. The transfer amplitude for bed topographic features aligned with the direction of ice flow approaches one as wavelength approaches infinity, but non-zero values of the basal sliding parameter $C^{0*}$ result in an additional peak in transfer amplitudes at intermediate wavelengths (as illustrated in Fig. 3, Gudmundsson, 2003). Transfer amplitudes approach zero at small wavelengths or as topographic features approach a flow-perpendicular alignment. Transfer function phase shift is also important, and results in a wavelength-dependent offset between bed features and their surface expression (as illustrated in Fig. 4.B).

Prior to taking 2D discrete Fourier transforms (DFTs, see for example Press et al., 2007) of bed DEMs, we first shift each bed DEM to have zero mean elevation. We do not detrend bed DEMs, as that is not consistent with the zeroth-order bed conditions.

We then mirror each bed DEM in all directions by connecting east-west reversed copies of each DEM to the east and west sides of itself, connecting north-south reversed copies of each DEM to the north and south sides of itself, and connecting north-south and east-west reversed copies of each DEM to all corners of itself. Next we apply a cosine taper such that all elevations along the edges of each mirrored bed DEM are zero, and the original domain in the center is unaffected. These processing steps are taken to minimize edge effects (Perron et al., 2008). We then take 2D DFTs of the mirrored bed DEMs, and use the transfer functions to obtain predicted ice surface elevation $Z'$ as:

$$Z'(x,y) = Z^0(x,y) + \mathcal{F}_D^{-1}\left(\hat{T}_B(k_x,k_y)\hat{B}(k_x,k_y) + \hat{T}_C(k_x,k_y)\hat{C}(k_x,k_y)\right) \tag{6}$$

where $B$ is the zero-mean bed elevation, $Z^0$ is the zeroth-order ice surface (with average elevation $H$ and slope $\alpha$, obtained by a plane fit to the ice surface DEM $Z$), and $\mathcal{F}_D^{-1}$ represents the inverse 2D DFT. We then trim enough space from the edges of each predicted surface so that we are only considering a region that will not contain any edge effects.

### 2.3.3 Ice viscosity and basal sliding estimation

Two important and poorly constrained parameters in our bed topography transfer method are ice viscosity $\eta$ and basal sliding parameter $C^*(x,y)$. One possible application of the transfer functions is to invert for these parameters as a function of space from observed surface and bed DEMs (Raymond and Gudmundsson, 2009). We do not take this approach here as our primary focus is an assessment of how well ice surface topography can be explained by transfer of basal conditions. However, we do need to choose values for $\eta$ and $C^*(x,y)$ (or at least a uniform value of $C^*(x,y) = C^{0*}$).

We examine the importance of spatial variations in $C^*(x,y)$ by comparing the predicted ice surface over Gaussian $B(x,y)$ and $C^*(x,y)$ perturbations with 2 km standard deviations and 200 m height or 200 $C^*$ amplitude, using ice flow parameters from region R1 (Fig. 1.A, Table 1) with $\eta = 10^{14}$ Pa s (and with $C^*(x,y) = C^{0*} = 10$ for the Gaussian bed topography test case); the results are shown in Fig. 4. Bed topography perturbations on the order of 200 m occur commonly (Morlighem et al., 2017a), but inferred slip ratios away from ice steams in the western Greenland ablation zone are typically less than $\sim 10$ (Morlighem et al., 2013; MacGregor et al., 2016). Thus Fig. 4.C indicates that in these flow conditions, unless there are exceptionally large $C^*(x,y)$ spatial perturbations (on the order of 1000), the ice surface expression from $C^*(x,y)$ perturbations will be of much smaller amplitude and more disperse than the surface expressions that can arise from reasonable amplitude bed topography. Accordingly, we assume spatially constant $C^*$ (so $C^*(x,y) = C^{0*}$ at all locations) for all of our analysis, so that we only need to choose a single value of $C^{0*}$ in each region.

We next assess the uniqueness with which $C^{0*}$ and $\eta$ can be inverted for using the transfer functions, by minimizing misfit (defining misfit for DEMs of size $m \times n$ as: $\frac{1}{nm}\sum_{x=x_1}^{x_n}\sum_{y=y_1}^{y_m}|Z(x,y) - Z'(x,y)|$) between observed and bed topography transfer predicted ice surfaces. Example inversion results are shown in Fig. 5. Over our seven study regions of the Greenland Ice Sheet ablation zone (Fig. 1.A, Table 1), the values of viscosity that produce best fits between predicted and observed ice surfaces are within half an order of magnitude of $10^{14}$ Pa s. For all further analysis we fix the value of $\eta$ to $10^{14}$ Pa s, which is within the range of ice viscosity estimates (Cuffey and Paterson, 2010).

The best fitting values of $C^{0*}$ in our study regions range between 6 and 35, and are often not very tightly constrained. Some of these values are significantly higher than other ice sheet ablation zone estimates (Morlighem et al., 2013; MacGregor et al., 2016). Such anomalously high $C^{0*}$ values are not unexpected, for at least two reasons. The first is poor effective bed DEM resolution in some regions, which could result in transfer amplitudes (and thus basal sliding) needing to be artificially high

to produce observed surface topographic relief. In regions with lower mean bed DEM error, our inversions result in lower values of sliding (Table 1). The second reason is that the Newtonian rheology used to derive the analytical transfer functions produces generally lower transfer amplitudes than are found with a more realistic (power law) ice rheology (Raymond and Gudmundsson, 2011), so artificially high basal sliding values are needed to produce observed transfer amplitudes. Except where otherwise noted we therefore set $C^{0*} = 10$, which is consistent with the linear transfer functions and likely over-predicts

true average slip ratios. Much of our analysis will focus on the wavelengths at which bed topography transfer peaks, which are relatively insensitive to the value of $C^{0*}$. This is because $C^{0*}$ affects transfer peak amplitude but not wavelengths (as can be seen in Fig. 3.C); for parameters representative of our study regions a significant peak is still predicted as long as $C^{0*} >\sim 2$.

### 2.3.4  Bed DEM error analysis

A significant source of error in our bed topography transfer function method is bed DEM accuracy. The BedMachine v3 bed

DEM has a corresponding potential error map which represents the uncertainty in bed elevations (shown in Fig. 2.A). This uncertainty primarily reflects poor radar transect coverage and generally increases with distance from the nearest radar data, but also depends on uncertainty in other data used for mass conservation modeling (Morlighem et al., 2014, 2017a). We use many randomly generated possible error configurations to quantitatively bound the variation in our ice surface topography predictions that is allowed by bed DEM uncertainty. This allows us to assess the robustness of our surface predictions, and to

examine the bed DEM accuracy needed for reasonable predictions.

We pseudo-randomly generate 100 possible error configurations for each of 100 different bandpass filter wavelengths $\lambda_n$ (where $\lambda_n$ spans the range of wavelengths resolvable in each domain). Each error configuration is created from a different pseudo-random complex wavenumber matrix. The matrices have the same dimensions as the bed DEM, symmetric real components, and anti-symmetric imaginary components. Each wavenumber matrix is multiplied with a frequency domain Gaussian

bandpass filter centered at frequency $\frac{1}{\lambda_n}$. An inverse DFT is taken of each wavenumber matrix to create an error surface containing primarily topographic wavelengths near $\lambda_n$. Each error surface is then scaled so that all values vary between -1 and 1, and multiplied by the bed DEM potential error map to generate a possible error configuration. We add each error configuration to the bed DEM and use bed topography transfer functions to predict the ice surface over each resulting error-injected bed DEM.

### 30   2.3.5  Observed admittance of ice surface/bed topography

We wish to evaluate how well the observed frequency domain empirical admittance of bed topographic features to the ice surface corresponds to predicted transfer amplitudes, to determine the wavelengths at which observed ice surface topographic amplitudes are consistent with predicted bed topography transfer (over given bed DEMs). We can calculate the frequency

domain empirical admittance of bed topography to ice surface topography as $\hat{Y}(k_x, k_y) = \hat{Z}(k_x, k_y)/\hat{B}(k_x, k_y)$. If ice surface topography was only caused by bed topography transfer, then empirical admittance $\hat{Y}$ should closely correspond to predicted bed topography transfer amplitudes. However, due to the noise present in 2D empirical admittance computations, interpreting $\hat{Y}(k_x, k_y)$ directly is challenging. We thus employ two methods to estimate average empirical 1D admittance $\hat{Y}(k)$, which we can then compare to predicted transfer amplitudes.

One method to estimate 1D empirical admittance involves binning and averaging absolute wavenumber components from 2D DFTs. We first take 2D discrete Fourier transforms (DFTs) of mirrored and tapered ice surface and bed DEMs (as described in Section 2.3.2), then calculate the complex magnitudes of all values in these DFTs to yield 2D surface and bed amplitude spectra. We then bin and average all points in each amplitude spectra by absolute wavenumber to obtain 1D surface and bed amplitude spectra. We lastly divide binned 1D surface amplitude spectra by binned 1D bed spectra. This method considers both ice-flow-parallel and non-ice-flow-parallel topographic wavelengths, which could decrease the resulting admittance relative to admittance expected purely in the ice flow direction (since transfer should be highest for topographic wavelengths aligned in this direction).

The second method to estimate 1D empirical admittance involves binning and averaging 1D amplitude spectra from multiple ice flowlines. We first interpolate bed and surface elevations along a series of offset near-parallel ice flowlines. We mirror and taper each flowline elevation profile, then take DFTs of each profile to obtain a series of 1D surface and bed amplitude spectra. We next bin and average each 1D amplitude spectrum by wavenumber, and average these spectra between all profiles of the same type (surface or bed). We lastly divide binned and averaged 1D ice surface amplitude spectra by binned and averaged 1D bed amplitude spectra. This method thus avoids the non-ice-flow-parallel muting effect from the first method, but does not account for the effects of surrounding 2D topography on each flowline (as discussed in Section 2.3.2).

## 2.4 Supraglacial meltwater routing and thermal-fluvial incision

### 2.4.1 Mechanics of fluvial incision

In terrestrial settings, bedrock fluvial incision is often modeled by the "stream power" law (Howard and Kerby, 1983; Seidl and Dietrich, 1992). This model can be combined with another semi-empirical relation Hack's law (Hack, 1957), relating downstream distance to accumulated flow area. This permits prediction of surface lowering by fluvial erosion $E$ of the substrate at point $s$ along a stream channel downstream of a drainage divide at time $t$

$$E(s,t) = K(s,t)A(s,t)^m \left| \frac{\partial Z(s,t)}{\partial s} \right|^n, \tag{7}$$

where $A(s,t)$ is accumulated drainage area, $K(s,t)$ is an experimentally determined erodibility coefficient that may vary in space and time, $m$ and $n$ are empirically determined exponents, and $Z(s,t)$ is channel elevation. This model, combined with models for tectonic uplift or hillslope creep, well-predicts large-scale features of many fluvially-dominated terrestrial landscapes. Commonly observed concave-up longitudinal stream elevation profiles and negative slope-drainage area trends are generally interpreted in the context of equation 7, which then may inverted for tectonics and climate, or used to constrain

substrate properties such as erodibility $K$ (Gilbert, 1877; Whipple and Tucker, 1999; Montgomery, 2001). Convexities such as those induced by base level changes, non-uniform uplift, and variable climate or substrate properties propagate upstream as kinematic waves (Whipple and Tucker, 1999; Royden and Perron, 2013; OHara et al., submitted 2018).

In supraglacial environments fluvial incision occurs by melting, and an analog of the stream power law may be derived with $n = 1$ (Karlstrom and Yang, 2016). Exponent $m$ is dependent upon the relation between water flux and accumulated drainage area and the relation between channel width and water flux, and has been estimated at between 0.7-0.9 for supraglacial streams (Karlstrom and Yang, 2016). If surface motions introduced by ice advection (analogous to unsteady and non-uniform uplift) are accounted for, fluvially-dominated supraglacial stream profiles with fixed terminal elevations (such as supraglacial lakes) should still approach a concave-up configuration if thermal-fluvial erosion outpaces ice advection (Karlstrom and Yang, 2016). Equation 7 also implies that for fluvially-dominated stream profiles without fixed terminal elevations (such as those flowing into moulins), convexities can progressively propagate upstream from the moulin causing persistent transient topography. Indeed, convexities at various scales are readily visible in supraglacial stream elevation profiles (see supplement), but these deviations from idealized longitudinal profiles could arise from other processes as well. Spatially varying background ice flow, kinematic waves transmitting uplift or erosion transients (such as from unsteady surface melting or supraglacial lake drainage (Hoffman et al., 2011)), transient surface waves caused by ice flux variations (van de Wal and Oerlemans, 1995), and/or deviations of the local ice velocity vector from the direction of stream flow (such as from stream meanders, e.g., Karlstrom et al., 2013) could all generate convexities in fluvially-dominated supraglacial stream profiles. Alternately, if thermal-fluvial incision is slow enough relative to ice advection and/or other surface processes, stream profiles might not be primarily controlled by fluvial incision, and instead would conform to the shape of the surrounding topography that is controlled by other processes.

Modeling the dynamic interaction between thermal-fluvial incision and ice advection is beyond the scope of this work, and such modeling would still be limited by the resolution of current bed DEMs that affects our transfer function implementation (e.g., Sections 2.1, 2.3.4, and 3.1). We thus instead employ two empirical approaches to search for signatures of IDC-scale landscape modification by thermal-fluvial incision, and to quantify the observed pattern of supraglacial stream networks in relation to bed topography transfer. The first approach is to compare slope versus accumulated drainage/flow area relations, a traditional terrestrial landscape metric (Gilbert, 1877; Whipple and Tucker, 1999; Montgomery, 2001), between real supraglacial stream networks and synthetic flow networks calculated on bed topography transfer predicted surfaces (described in Section 2.4.3). The second approach is to use two stream conformity metrics to quantify how well supraglacial stream network geometry is explained by the surrounding ice surface topography filtered at various wavelength thresholds (described in Section 2.4.4).

### 2.4.2 Supraglacial stream network and synthetic flow network extraction

We use satellite imagery, DEMs, and flow-routing algorithms to extract supraglacial stream networks from seven regions of the western Greenland Ice Sheet ablation zone (Karlstrom and Yang, 2016; Yang and Smith, 2016). Satellite imagery is used to identify moulins by hand, which are treated as water sinks. We then use flow-routing to calculate accumulated flow/drainage area patterns on the surface (as shown in the example stream network in Fig. 6.A). We use the D8 (steepest descent) flow-routing algorithm with channel area threshold set to maximize agreement with visible stream channels, between 8000 and

30000 m$^2$ depending upon region. In general, flow-routing is an imperfect means of finding real stream channels, especially on a relatively flat landscape such as the Greenland Ice Sheet. DEM resolution is not high enough to resolve narrow ($<$ $\sim$2 m wide) supraglacial stream channels, so such streams may be missed by flow-routing, particularly those that are not aligned with the steepest descent direction (Smith et al., 2015; Yang et al., 2015). However, most streams found via our flow-routing method agree with visually identified stream channels (Yang and Smith, 2016).

Bed topography transfer provides a way of constructing synthetic flow networks to examine how meltwater would route in the absence of supraglacial thermal-fluvial incision, since incision is not accounted for by the transfer functions. We use transfer functions to predict the ice surface over bed DEMs, then place artificial moulins as water sinks at the base of large surface depressions and calculate synthetic flow networks numerically. These are not perfectly comparable with real supraglacial stream networks, since moulins also occur outside of depressions (Catania et al., 2008; Yang et al., 2015; Yang and Smith, 2016). We calculate these synthetic flow networks with the TopoToolbox (Schwanghart, 2014) D8 method, with channel area threshold set to 20000 m$^2$.

### 2.4.3 Supraglacial stream network slope and accumulated drainage area relations

Our first approach for quantifying controls on meltwater routing comes from the hypothesis that bed topography transfer can explain supraglacial stream longitudinal elevation profiles, without appealing to significant landscape shaping by thermal-fluvial incision. Although modeling the transient competition between ice flow over bed topography and thermal-fluvial incision is outside the scope of the present work, if bed topography transfer is the dominant process then slope-drainage relations on synthetic flow networks will match those from observed supraglacial stream networks. If instead supraglacial stream incision is a primary control on ice surface topography at km scales, the interplay between thermal-fluvial erosion and ice flow will set the longitudinal profiles of streams and the relationship between slope and accumulated drainage area of observed stream networks may consistently differ from synthetic flow networks.

We compare local channel slope to local accumulated upstream flow/drainage area at all points in each stream network. Prior to doing this we smooth all stream longitudinal profiles to remove small-scale slope variations. Profile smoothing is done by first breaking each stream network into multiple separate stream profiles, discarding all profiles less than 800 m long, then twice applying a moving average filter with a span of 200 m (analogous to a lowpass filter) to each remaining profile, and finally trimming 100 m from both ends of each profile to remove smoothing-induced edge effects. We then calculate stream longitudinal slopes with a second order centered finite difference stencil. There is a large scatter in the resulting slope versus drainage area relations, so for each stream network we divide data points into logarithmically spaced area bins and calculate the mean and standard deviation of slopes in each bin (Montgomery, 2001; Warren et al., 2004).

### 2.4.4 Supraglacial stream network topographic conformity

Our second approach for quantifying controls on meltwater routing is to implement two measures of stream network conformity to surrounding ice surface topography, as in Black et al. (2017). This approach assesses the degree to which stream patterns are "explained by" the current configuration of surrounding ice surface topography at various wavelengths. For a given stream/flow

network projected onto a given DEM, percent downhill ($\%d$) measures the percentage of channel length over which water forced along the channels would be flowing downhill, and conformity factor ($\Lambda$) measures the mean deviation of channel pathways from the local direction of steepest descent on the DEM surface (as illustrated in Fig. 6.B). We low-pass filter ice surface DEMs using a series of decreasing cutoff and taper wavelengths, then calculate $\%d$ and $\Lambda$ by projecting stream networks onto each filtered surface (as illustrated in Fig. 6.B). We note that applying these conformity metrics to stream networks calculated with flow-routing may result in a bias towards artificially high conformity, since as mentioned in Section 2.4.2 flow-routing on imperfect DEMs may miss some narrow stream channels that are are not aligned with the steepest descent direction on the ice surface. However, our flow-routing is done on sufficiently high-resolution DEMs to correctly capture the majority of observed stream network structures.

As filter cutoff wavelength decreases, both conformity metrics will increase if stream network geometry is controlled by the progressively shorter wavelengths of topography that are being included (Black et al., 2017). Given a DEM with high enough resolution to resolve all stream channels, as filter cutoff wavelength approaches zero $\%d$ should generally increase and approach $100\%$ since water does not flow uphill (except at vertical scales smaller than water flow depth). Similarly, $\Lambda$ should generally increase and approach 1, since water should generally flow in the direction of steepest descent. Stream network structure might depend on particular wavelengths of topography for a variety of reasons, for example if those wavelengths encompass topographic features that predate stream formation and thus contributed to the routing of the stream channels when they formed. Alternately, stream networks might not perfectly conform to the surrounding longer wavelength topography if fluvial meanders have shifted channels away from the background direction of steepest descent, or if the surrounding topography has been modified post stream-incision by processes such as ice advection (e.g., for tectonic processes, Black et al., 2017; Wegmann et al., 2007). We do not focus on why stream network conformity might be imperfect at any given wavelength, but instead use the conformity metrics to indicate what topographic wavelengths are important for explaining current supraglacial meltwater routing.

To calculate the two conformity metrics, we apply pre-processing steps as described in Section 2.3.2 to minimize edge effects, then low-pass filter ice surface DEMs using one-sided Gaussian filters. We then project flow networks (as computed on the unfiltered DEMs) onto each filtered surface (as illustrated in Fig. 6). We calculate $\%d$ as the percent of discrete locations along stream channels that are higher in elevation than the next downstream location. To calculate $\Lambda$, at each discrete location along a stream we calculate the angle between the horizontal direction vector of the stream channel (the direction water is flowing) and the horizontal direction vector of steepest descent down the ice surface. $\Lambda$ is then given by the mean absolute value of the cosine of this angle at all discrete stream channel locations. Exact expressions for $\%d$ and $\Lambda$ are given in the supplement.

## 2.5 Predicting supraglacial topographic drainage basins and subglacial hydraulic flow pathways

Bed topography transfer functions provide a tool for examining the effects various multiple-year averaged ice flow parameters have on ice surface topography. To do this we first predict ice surface topography (as described in Section 2.3.2) in a given

region with different ice flow parameters. We can then explore the effects these changes in surface topography might have on both supraglacial and subglacial hydrology.

To examine potential changes in supraglacial hydrology, we delineate surface topographic drainage basins on the predicted ice surfaces. We do this using flow-routing (with all topographic local minima treated as water sinks, as described in Section 2.4.2) to identify topographic drainage basin divides, counting all edge terminating basins separately. Topographic basins will not exactly correspond to IDCs, since moulins fragment topographic basins and/or there could be places where streams have incised through topographic divides (Yang et al., 2015). In practice there is reasonable correspondence between topographic basins and IDCs if appropriate DEM processing is used (Yang and Smith, 2016), so this approach provides a reasonable indication of how IDC configuration and number density would vary with changing multiple-year averaged ice flow parameters.

To explore potential changes in subglacial hydrology that might arise from changing ice flow conditions, we model quasi-static water flow patterns under the predicted ice surfaces. We first calculate subglacial hydraulic potential $\phi_h$ as a function of relative bed elevation and ice thickness following Hewitt (2011)

$$\phi_h(x,y) = \rho_w g B(x,y) + \rho_i g H(x,y) \left( \frac{P_w}{P_i} \right) \tag{8}$$

where $\frac{P_w}{P_i}$ is the ratio of basal water pressure to ice overburden pressure. Significant spatial and temporal variation in subglacial effective pressure $(P_i - P_w)$ under the Greenland Ice sheet has been measured, with basal water pressure ranging from less than half of ice overburden pressure to greater than ice overburden pressure (generally by only on the order of tens of bars, though brief pulses of much higher pressure have been recorded in some settings (Kavanaugh and Clarke, 2000)); this effective pressure variation is related to time of year, time of day, basal sliding velocity, and location within subglacial drainage networks (Ryser et al., 2014a; Andrews et al., 2014; Hoffman et al., 2016). Here we assume basal water pressure is equal to ice overburden pressure everywhere, which provides a reasonable upper-bound estimate of the direct impact ice surface topography could have on subglacial hydraulic potential. Subglacial water is often modeled as flowing down gradients in hydraulic potential (Hewitt, 2011; Wright et al., 2016). We thus apply flow-routing to the hydraulic potential fields to determine water flow paths and create accumulated flow/drainage area maps. We first fill sinks (local minima) in the hydraulic potential field in order to force all water to flow out of the domain. We then apply a multi-direction flow-routing algorithm from TopoToolbox (Schwanghart, 2014) since this produces more realistic flow pathways than D8 flow-routing in low-gradient areas (Quinn et al., 1991). We cannot account for water flow into the domain from up-gradient regions with this approach, so the drainage areas we calculate are lower bounds. These simple calculations also do not account for many important factors influencing subglacial hydrology such as basal melting/freezing, permeability, and subglacial channelization (Rempel, 2009; Schoof, 2010; Sole et al., 2011; Werder et al., 2013; Chandler et al., 2013), nor do they account for variation of flow pathways on timescales that differ from ice flow changes. However, they provide a useful tool for exploring the sensitivity of subglacial hydrology to the perturbations in surface topography caused by changing multiple-year averaged ice flow parameters.

# 3 Results

## 3.1 Bed topography transfer

We use two methods (as described in Section 2.3.5) to calculate the empirical admittance of bed topography from BedMachine DEMs (Morlighem et al., 2017a, b)) to observed ice surface topography in our seven study regions on the western Greenland Ice Sheet ablation zone. Results are shown in Fig. 7. In all regions, both calculations of empirical admittance (Fig. 7.B and Fig. 7.C) correspond well to predicted bed topography transfer amplitudes (Fig. 7.A, the transfer functions are described in Section 2.3.1) at wavelengths $> \sim 1$ km. Notably, both calculations of empirical admittance generally exhibit peaks at wavelengths from $\sim$1-10 km, consistent with what would be predicted from bed topography transfer. However, in both calculations empirical admittance at wavelengths $< 1$ km is higher than would be predicted from bed topography transfer. We expect this is in part due to limited effective bed DEM resolution at these shorter wavelengths, and in part due to other processes creating short-wavelength surface topography (such as fluvial incision and crevassing). That there is good agreement between the transfer functions and both calculations of empirical admittance at wavelengths $> \sim 1$ km provides one piece of evidence that bed topography transfer is a dominant control on surface topography at these scales.

We then use the steady-state bed topography transfer functions to predict ice surface topography (as described in Section 2.3.2) in our seven study regions (Fig. 1.A, Table 1). Example results from region R1 are shown in (Fig. 8). In regions R1, R2, and R7 the transfer functions qualitatively well predict general IDC-scale ($\sim$1-10 km, consistent with our admittance calculations) features of the ice surface, such as large ridges and depressions (see Fig. 8.A,B and supplement). In regions R3, R4, R5, and R6 the transfer functions significantly "under-predict" surface topography by creating noticeably smoother surfaces than observed. We expect that this is primarily due to the limited effective bed DEM resolution in these regions, as discussed below.

To quantitatively evaluate the effectiveness of our ice surface topography predictions, we calculate mean misfit as a percentage of surface topographic relief in each study region. For a DEM of size $m \times n$ this misfit metric is expressed as: $\frac{1}{nm} \frac{100}{\text{Range}(Z)} \sum_{x=x_1}^{x_n} \sum_{y=y_1}^{y_m} |Z(x,y) - Z'(x,y)|$; we note this metric is not necessarily a comprehensive indicator of fit quality. Mean misfits for all study regions are shown in Table 1, and an example misfit map is shown in Fig. 8.C. Even in regions where bed topography transfer predictions qualitatively well produce km-scale surface topographic features misfit is still significant; mean misfit values are 9-14% of regional topographic relief. As discussed in Sections 2.3.1 and 2.3.2, there are many potential causes of such misfit: bed DEM error, the various assumptions made in deriving and implementing the transfer functions (such as assuming Newtonian ice rheology, linearity, and a steady-state limit), and/or unaccounted for processes such as fluvial incision and kinematic ice waves. We use the approach described in Section 2.3.4 to examine the potential effects of bed DEM error on ice surface predictions in our study regions. This can be significant, ranging from $\sim$40% to larger than 100% of regional ice surface relief depending upon the configuration and magnitude of DEM error. However, where bed DEMs are relatively accurate (generally less than $\sim$60-100 m potential error) these error effects are smaller than the regional ice surface relief (as shown in Fig. 8.D), indicating that large-scale features of surface predictions in these areas should be meaningful. Unfortunately potential bed DEM error is currently worse than 100 m over much of the Greenland Ice Sheet (Morlighem et al.

(2017a, b), Fig. 2), limiting the possible precision of surface predictions or inversions for parameters like basal sliding ($C^*$) in many regions.

Thus we have shown that, where bed DEMs are sufficiently accurate, bed topography transfer can explain IDC-scale ($\sim$1-10 km) ice surface amplitude spectra and IDC-scale ice surface topographic features. This provides verification that bed topography is a dominant control on IDC scale surface topography, though with insufficient resolution to directly quantify the significance of other processes like thermal-fluvial incision that are superimposed on the effects of bed topography.

## 3.2   Supraglacial stream network slope and accumulated drainage area relations

Supraglacial stream networks from our seven study areas (Fig. 1.A, Table 1) all exhibit negative slope versus drainage area relationships (thus positive concavity), as shown in Fig. 9. This is expected in a fluvially controlled landscape (as discussed in Section 2.4.3, or see Montgomery (2001)). However, negative slope-area relations can arise without fluvial incision in randomly generated DEMs (Schorghofer and Rothman, 2002), so in isolation this geomorphic metric is challenging to invert uniquely for process. We thus use control cases with no fluvial influence for comparison; these controls are synthetic flow networks created by artificially placing moulins on bed topography transfer predicted ice surfaces (as described in Section 2.4.2). The map-view structure of synthetic flow networks is not realistic, since the bed topography transfer predicted surfaces are very smooth and D-8 flow-routing then produces straight and parallel channels. However, in slope-drainage area space, synthetic flow networks in regions with qualitatively reasonable surface predictions (R1, R2, and R7) exhibit similar negative slope-area trends to the corresponding observed stream networks, as shown in Fig. 9. The slope area trends of regions R3, R4, and R5 are noticeably flatter than the corresponding observed stream networks. We expect this is mainly because limited effective bed DEM resolution results in under-predicted surface topography, on which all surface slopes deviate minimally from the regional background slope ($\alpha$).

In regions with more reliable surface predictions (R1, R2, and R7), synthetic and observed slope-area trends have similar slopes, as shown by the power-law fits in Fig. 9. There are deviations between observed stream networks and synthetic flow networks in regions R1, R2, and R7, but there is not a clear consistency in such deviations between these regions. Given the very large scatter inherent to slope-area relations (shown in Fig. 9 and discussed by Warren et al. (2004)) and the limitations of our surface predictions, it is difficult to say from this data if there are consistent differences between observed slope-area relationships and those calculated on bed topography transfer predicted surfaces that could indicate fluvial modification of stream longitudinal elevation profiles. Further study with better bed DEMs and more detailed ice flow modeling might tease out such fluvial signatures. However, our results are sufficient to show that given accurate enough bed DEMs, bed topography transfer alone can produce synthetic stream networks with longitudinal slope-area structure approximately similar to observed stream networks.

## 3.3   Supraglacial stream network topographic conformity

We calculate both stream network topographic conformity metrics (as described in Section 2.4.4) for supraglacial stream networks from our seven study regions (Fig. 1.A, Table 1); results are shown in Fig. 10. In all regions there are consistent

trends in both $\%d$ (percent downhill) and $\Lambda$ (conformity factor). At the longest wavelength cutoffs $\%d$ and $\Lambda$ are at their lowest regional values, and including shorter topographic wavelengths generally results in increases in both metrics. $\%d$ and $\Lambda$ plateau at values between $\sim 88 - 97\%$ and $\sim 0.75 - 0.81$ respectively. That these values plateau at less than the maximum respective values of 100 and 1 in real stream networks could be due to varying channel depths and/or DEM inaccuracy; we normalized the values of both metrics in Fig. 10 to better highlight how the metrics change from their plateau values as progressively longer topographic wavelengths are removed.

In all stream networks the most significant decreases in both $\%d$ and $\Lambda$ occur in bands of cutoff wavelengths roughly between 1 and 10 km. This indicates that these wavelengths of topography are the wavelengths that are most important for explaining the overall structure of supraglacial stream networks. These wavelength bands match the wavelengths at which predicted bed topography transfer is highest, and also where we find peak admittance between surface and bed DEMs (see Fig. 7). In particular, we note that the region where stream conformity is more affected by smaller wavelengths (solid red curves) would be expected to exhibit comparatively high bed topography transfer at these smaller wavelengths. The regions where stream conformity is less affected by smaller wavelengths (solid yellow, green, and purple curves) would be expected to exhibit comparatively low bed topography transfer at these wavelengths. Thus in all of our study regions the general routing of surface meltwater according to these conformity metrics is consistent with control by bed topography.

Our combined results thus demonstrate that given sufficiently accurate bed DEMs, bed topography transfer alone can reasonably well explain ablation zone IDC-scale ($\sim$1-10 km) ice surface topography and meltwater routing. This conclusion is supported by surface topographic admittance calculations, bed transfer predictions of surface topography, and three different geomorphological metrics of supraglacial stream network structure. This suggests that the effects of thermal-fluvial incision on IDC-scale supraglacial meltwater routing are secondary, superimposed on the dominant basal control of surface topography.

## 4 Discussion

### 4.1 Predicting supraglacial IDC evolution

Given moulin locations and ice flow conditions, our results imply that bed topography transfer should generally explain IDC configurations, such as the trend observed by Yang and Smith (2016) where average IDC area increases with increasing ice surface elevation/thickness. The bed topography transfer functions also provide a tool to perform a parameter study and predict IDC-scale surface topography under different multi-year averaged ice flow conditions. Even without predicting moulin locations, we can still use our methodology to examine the general response of surface topographic basins to changing ice flow conditions as described in Section 2.5. This is important since surface topography and IDC configuration could impact subglacial hydrology, as we discuss in the next section (Section. 4.2). Additionally, it is expected that the ablation zone of the Greenland Ice Sheet will move to higher elevations in coming years as global climate warms (Rae et al., 2012; Fettweis et al., 2013; Leeson et al., 2015). Given moulin locations, an approach similar to what we implement here could be used to obtain precise predictions of the of the spatial and temporal input of surface meltwater into moulins if combined with tools such as hydrographs (e.g., Smith et al., 2017).

The topographic basins associated with a predicted ice surface in different multiple-year averaged ice flow conditions are shown in Fig. 11. Variations from current ice flow parameters by factors of 1/2 and 2 illustrate parameter sensitivity, and are not based off of any predictions for how much each multi-year averaged parameter might change in a particular timescale. We also note that topographic basins will not exactly correspond to IDCs (Smith et al., 2015; Yang et al., 2015; Yang and Smith, 2016); for comparison we show IDC configurations obtained solely from satellite imagery by Yang and Smith (2016) in Fig. 11.A. Despite the visible differences between our bed topography transfer predicted topographic basin configuration and the observed IDC configuration, the overall basin and IDC number densities are similar. This is consistent with results from Yang and Smith (2016) showing that surface topography roughly predicts IDC configurations. We thus expect that changes in topographic basin density predicted with changing ice flow conditions should generally correspond to changes in IDC density. Topographic basin density is not significantly affected by factor-of-four increases in ice surface slope $\alpha$ (Fig. 11.B7-B8, from 0.12-0.10 basins/km$^3$) or ice surface velocity $U$ (Fig. 11.B3-B4, from 0.10-0.11 basins/km$^3$). However, topographic basin density decreases appreciably with factor-of-four increases in ice thickness $H$ (Fig. 11.B1-B2, from 0.16-0.08 basins/km$^3$), and increases appreciably with factor-of-four increases in basal sliding $C^{0*}$ (Fig. 11.B5-B6, from 0.08-0.18 basins/km$^3$). Our analysis thus indicates that ice surface topographic basin density in the Greenland Ice Sheet ablation zone could be significantly affected by changes in multi-year averaged ice thickness or basal sliding.

As discussed in Section 2.3.1, the timescale over which the ice sheet surface approaches 95% of its steady state configuration in response to a basal perturbation is on the order of 3-60 years depending upon perturbation wavelength, so the results here (and in the following Section 4.2) should be interpreted as predicting multiple-year averaged ice surface configurations. Minimal adjustment to changing ice flow or basal sliding conditions is predicted on shorter seasonal timescales, although increasingly high temporal resolution observations could motivate such shorter timescale modeling in the future.

## 4.2   Potential coupling between ice surface topography and subglacial hydrology

We have shown in Section 4.1 that changing ice flow conditions should result in changing ice surface topography and supraglacial IDC configuration (see Fig. 11); we can now explore and speculate upon how such changes might affect subglacial hydrology and/or basal sliding.

Perturbations to surface topography could have direct impacts on subglacial hydraulic potential, and thus on subglacial water flow pathways. We calculate such pathways as described in Section 2.5; the results are shown in Fig. 12. The predicted variations in subglacial meltwater flow patterns are subtle, but there is some change in all cases. This is most visible where the configuration of high-flow-area paths changes, as can be seen near the center of the study region between Fig. 12.B5 and Fig. 12.B6. The threshold flow-area we use to calculate areal percentages in Fig. 12 ($5 \times 10^6$ m$^2$) was chosen to highlight pathways of high relative flow area. Subglacial channelization (discussed more later in this section) should occur preferentially around such pathways, since water flux should generally increase with increasing flow area (Hewitt, 2011; Wright et al., 2016). Doubling $C^{0*}$ or $\alpha$ slightly increases the percent of the study region covered by such higher-flow pathways, while doubling $U$ or $H$ has the opposite effect. The magnitude of these changes is generally less than around 20% of the baseline areal coverage for any chosen flow-area threshold. Dynamic subglacial hydrology models (such as Schoof, 2010 or Werder et al., 2013) are

needed to more completely assess the potential impacts of these changes. However, our results indicate that unless any of the multiple-year averaged ice flow parameters changes by more than a factor of two, the effects (that are directly caused by perturbations in surface topography) such changes will have on subglacial hydraulic pathways are likely to be subtle.

Our calculations suggest that the more important influence of ice surface topography on subglacial hydrology may be from the dispersion of surface meltwater input caused by changing surface topographic basin (or IDC) number density (as shown in Fig. 11). For a given melt production rate, if topographic basin density increases then meltwater input to the subglacial environment will be dispersed among more moulins, up to the point at which some basins become small enough that they fill and overtop without building up enough water pressure to generate moulins through hydrofracturing (Banwell et al., 2012, 2016). This dispersion of moulin water input could impact subglacial hydrology in several ways. If such dispersion results in average subglacial water pressure increases due to less effective or slower development of subglacial channels, this could lead to lower average basal effective stresses and increased basal sliding (Werder et al., 2013; Banwell et al., 2016; Hoffman et al., 2018). Alternately, if subglacial channelization happens rapidly regardless of meltwater input rate, then the dispersion of meltwater input may not be particularly significant or may result in more effective subglacial channel networks (Banwell et al., 2016). The extent to which subglacial channelization occurs is debated (Meierbachtol et al., 2013), but some subglacial channelization may occur on timescales of hours to days with continuing evolution over the length of melt seasons, and in the Greenland Ice Sheet ablation zone moulin meltwater input is a significant source of basal water affecting this subglacial drainage development (Schoof, 2010; Sole et al., 2011; Werder et al., 2013; Chandler et al., 2013). Of course, the total amount and timing of surface meltwater flux will also change if the annual surface energy budget varies (Cuffey and Paterson, 2010; Ahlstrøm et al., 2017), if the average albedo of IDCs varies (Leeson et al., 2015), or if partitioning between slow (porous snow/weathering crust flow, firn aquifer) and fast (stream channel) pathways varies (e.g., Karlstrom et al., 2014; Cooper et al., 2018; Yang et al., 2018). We see including such effects in glacial surface models as a promising avenue for future research.

Basal sliding is the parameter that generally has the most significant effect on surface topographic basin density (as shown in Fig. 11). Basal sliding can change significantly over timescales from hours to years, and often has strong seasonal cycles (Selmes et al., 2011; Sole et al., 2011; Chandler et al., 2013; Shannon et al., 2013). As discussed in Sections 2.3.2 and 2.3.3, the long term averaged basal sliding parameter we assume in implementing basal transfer functions may not directly relate to the real seasonally varying values of basal slip ratio (Tedstone et al., 2014). However, the effects from relative changes in basal sliding that our methods predict should be more robust, and it is reasonable to expect that persistent changes in basal sliding during melt seasons and/or in the length of melt seasons could have effects on surface topography that are analogous to this multi-year averaged basal sliding parameter. Our results thus indicate that there are potential feedbacks wherein changes in multi-year averaged basal sliding affect surface IDC configurations, which could in turn affect subglacial hydrology and basal sliding.

## 4.3 Thermal-fluvial incision on sub-IDC scales

Our analysis suggests that the influence thermal-fluvial incision has on large (> 1 km) scale surface topography and stream network structures must be secondary and superimposed on the dominant influence of bed topography. However, empirical

admittance calculations show that other influences on ice surface topography could become more significant at scales $< \sim 1$ km (see Fig. 7). We expect fluvial incision to be a primary influence on surface topography and meltwater routing pathways at these scales. Models that couple transient ice flow over rough bed topography to a surface energy balance, along with accurate bed DEMs, will be necessary to quantitatively constrain the influence of thermal-fluvial incision on ice surface topography and meltwater routing. Such models are also required to address observed supraglacial channel network coarsening (time evolution of channel density, (e.g., Yang and Smith, 2016)), and to establish how diurnally and seasonally varying melt rates are imprinted on stream networks.

From the standpoint of predicting Greenland Ice Sheet-wide hydrology, our work may simplify future modeling efforts. If thermal-fluvial incision does not significantly modify the ice surface at IDC-scales, as our results suggest, supraglacial stream incision would not need to be fully coupled with ice sheet models in order to predict meltwater routing and the larger-scale evolution of ice surface topography over long timescales. Future work towards this goal should focus on better determining bed elevations and predicting moulin formation (Joughin et al., 2013; Young et al., 2018).

## 5 Conclusions

Understanding the processes that govern surface meltwater routing on the Greenland Ice Sheet, and how this meltwater routing might change with changing climate or ice flow conditions, is important for understanding and predicting subglacial hydrology and ice sheet evolution. We implement linear transfer functions that predict the ice surface over rough bed topography in multiple 2D regions of the western Greenland Ice Sheet ablation zone. We verify that bed topography transfer alone, in the steady state limit, can largely explain $\sim$1-10 km wavelength ice surface topography under a range of ice flow conditions, given sufficient quality bed DEMs.

We then apply flow-routing to extract supraglacial flow networks from observed ice surface DEMs and from bed topography transfer predicted ice surfaces. We quantify stream network conformity to surrounding topography and estimate the relation between supraglacial channel slope and accumulated drainage area. These metrics are consistent with the inference that transfer of bed topography to the surface is the dominant process controlling general IDC-scale ($\sim$1-10 km) supraglacial meltwater routing on the Greenland Ice Sheet ablation zone.

Finally, we conduct a parameter sensitivity study to predict the adjustment of surface topography, supraglacial IDCs, and subglacial hydraulic potential that would occur in response to changing multi-year averaged ice flow conditions in a representative western Greenland site. We show that the surface topography perturbations caused by changing ice flow can have direct effects on subglacial hydraulic pathways. However, the more significant impact on subglacial hydrology may result from the increasing number density of surface IDCs, and the corresponding dispersion of englacial/subglacial surface meltwater input, that we show would be caused by decreasing ice thickness or increasing multi-year averaged basal sliding. This suggests a possible coupling between surface IDC configuration, subglacial hydrology, and basal sliding efficacy.

*Code and data availability.* All codes and data produced by the authors available upon request. Ice surface DEMs are from SETSM Arc­ticDEM 2-10 m resolution mosaics (ArcticDEM, 2017). Bed DEMs are from Icebridge BedMachine (Morlighem et al., 2014, 2015). Ice surface velocity data is from MEaSUREs (Joughin et al., 2010b, a). 2015 Melt data is from RACMO 2.3p2 (Noel et al., 2015).

*Author contributions.* Josh Crozier implemented most modeling and data analysis with input from Leif Karlstrom and Kang Yang; the
5   extraction of observed supraglacial stream networks was done by Kang Yang. Josh Crozier wrote the manuscript with input from Leif Karlstrom and Kang Yang. Leif Karlstrom, Kang Yang, and Josh Crozier conceived of the study.

*Competing interests.* No competing interests are present.

*Acknowledgements.* We thank Colin Meyer, Dan O'Hara, and Alan Rempel for their input and discussions. We thank two anonymous reviewers for their constructive comments. Leif Karlstrom acknowledges funding from NASA award NNX16AQ56G. Kang Yang acknowl­
10   edges support from the National Natural Science Foundation of China (41501452) and the Fundamental Research Funds for the Central Universities.

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

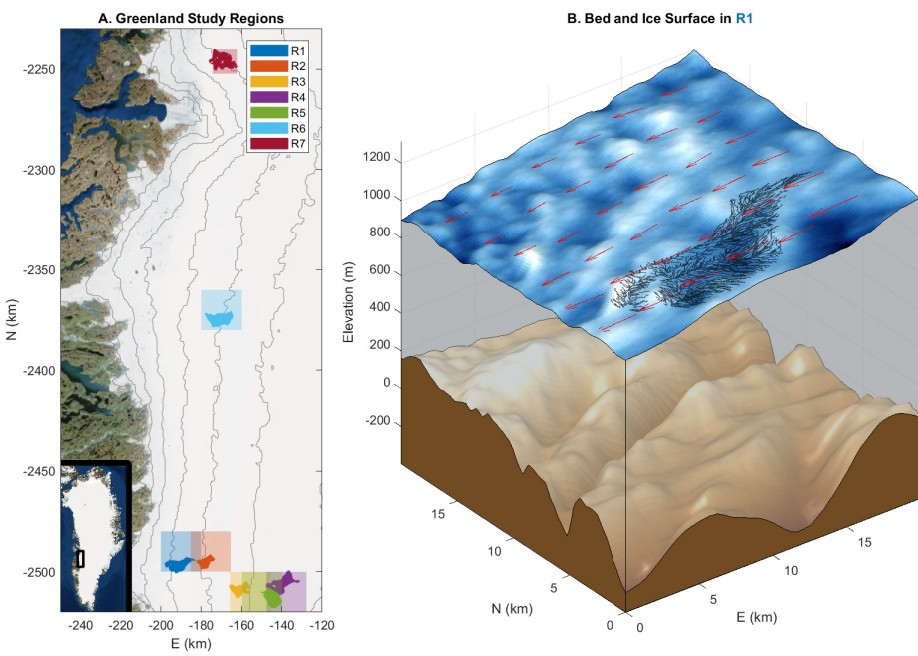

**Figure 1.** (A) Study IDCs (solid colored patches) of western Greenland with bounding boxes (semi-transparent squares) indicating corresponding domain used for bed transfer and admittance calculations. Black 200 m elevation contours are from BedMachine/GIMP (Howat et al., 2014; Morlighem et al., 2017a). Imagery is from ArcGIS ERSI world imagery basemap. Information on all regions is shown in Table 1. (B) Ice surface and bed elevations (from BedMachine/GIMP) in study region R1, with stream channels from study drainage network shown in black and velocity field shown by red arrows.

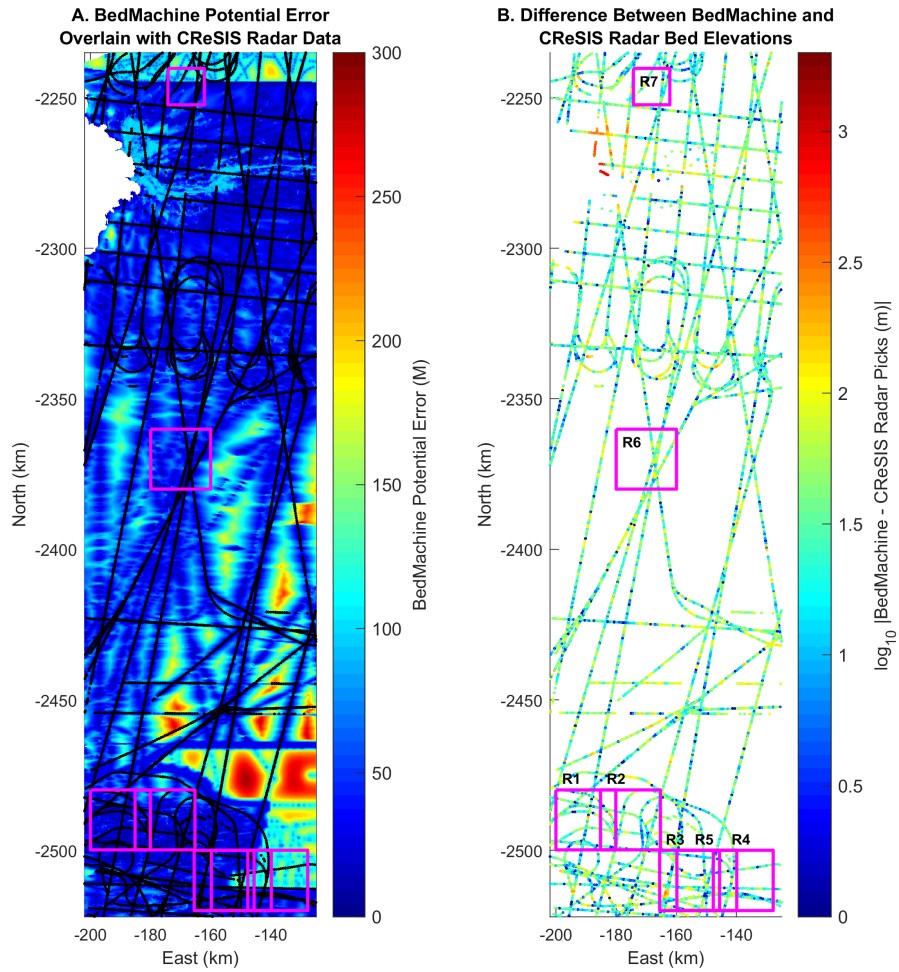

**Figure 2.** (A) BedMachine v3 potential bed elevation error overlain by CReSIS radar bed elevation picks (Morlighem et al., 2017a, b; CReSIS, 2016). BedMachine includes radar data from other sources not shown here, and is also constrained with mass conservation modeling over most of the region shown. BedMachine error generally decreases where elevations are better constrained by radar data. Our study regions (magenta) encompass a broad range of bed DEM quality (Fig. 1.A, Table 1). (B) Difference between bed elevations from BedMachine and CReSIS radar picks (only including radar picks marked as good quality). In many regions there is appreciable scatter in radar elevation picks and significant error in the derived bed DEM; we examine the impact this uncertainty has on ice surface predictions.

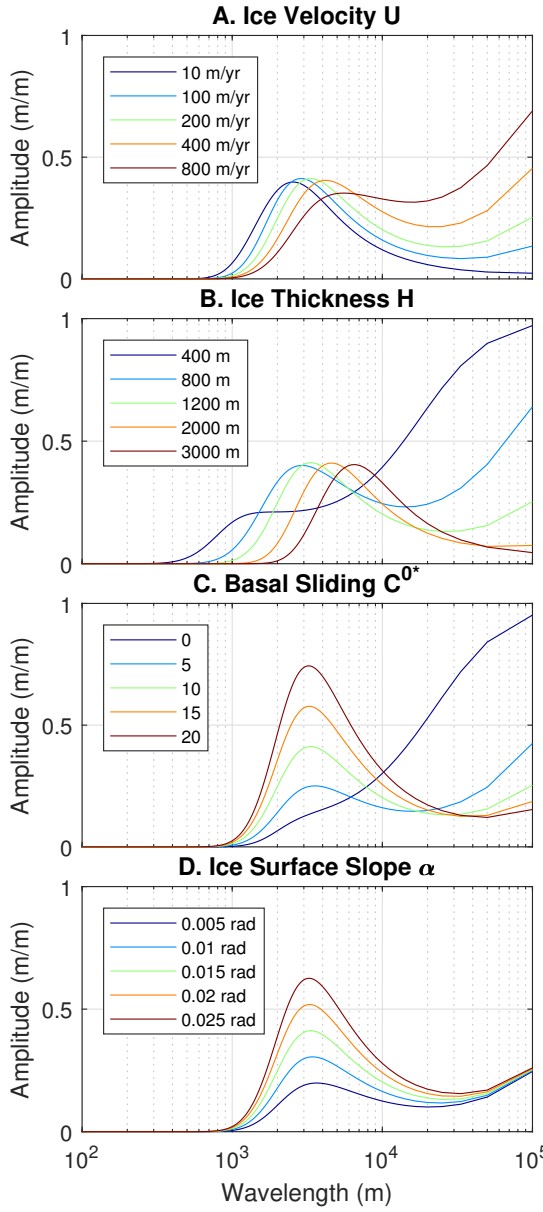

**Figure 3.** (A-D) Bed topography transfer function amplitudes in the ice flow direction (along an ice flowline). In all plots the parameters not otherwise indicated are: $U = 200$ m/yr, $H = 1200$ m, $C^{0*} = 10$, $\alpha = 0.015$ radians, and $\eta = 10^{14}$ Pa s. The spread of plotted parameters broadly encompasses the range of parameters found in our study regions (Fig. 1.A, Table 1). Transfer amplitude peaks between around 1-10 km for a wide range of parameters.

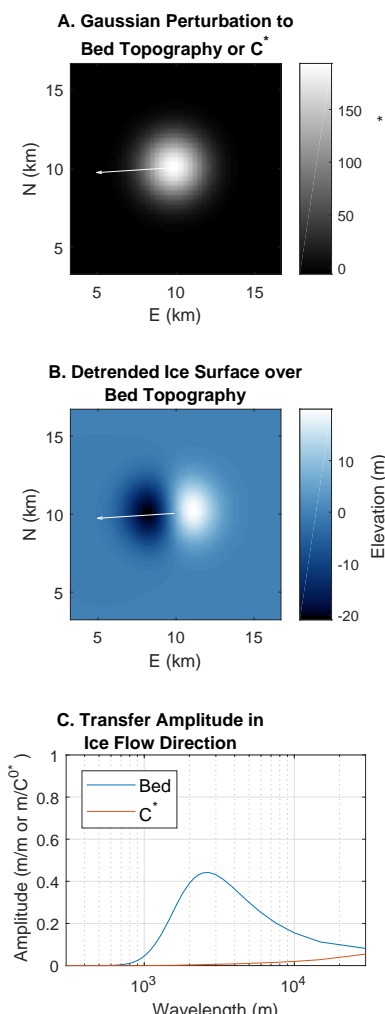

**Figure 4.** Illustration of steady state basal transfer for ice flow parameters representative of the western Greenland ablation zone. Ice flow parameters are from region R1 (Fig. 1.A, Table 1) with $\eta = 10^{14}$ Pa s. (A) Gaussian bed topography or basal sliding perturbation. (B) Detrended predicted ice surface over the Gaussian bed topography perturbation (with $C^{0*} = 10$). White arrows in plots A and B indicate the ice flow direction. (C) Transfer amplitudes in the ice flow direction (along a flowline) for bed topography and basal sliding $C^*$ perturbations. The transfer functions also have important phase components not shown here (see Gudmundsson (2003)). With these flow parameters, surface topography created from basal sliding perturbations should generally be of much lower amplitude than surface topography created from bed topography.

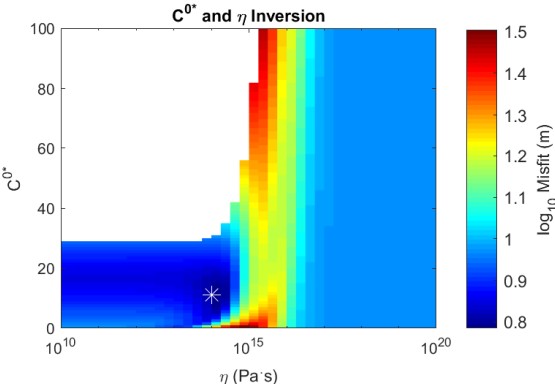

**Figure 5.** Misfit minimization for $\eta$ and $C^{0*}$ between the ice surface DEM and bed topography transfer predicted ice surfaces in study region R1 (Fig. 1.A, Table 1). White star indicates the location of minimum misfit, at $C^{0*} = 10$ and $\eta = 10^{14}$. Blank plot area is the region where parameters are nonphysical (resulting in transfer amplitudes $> 1$).

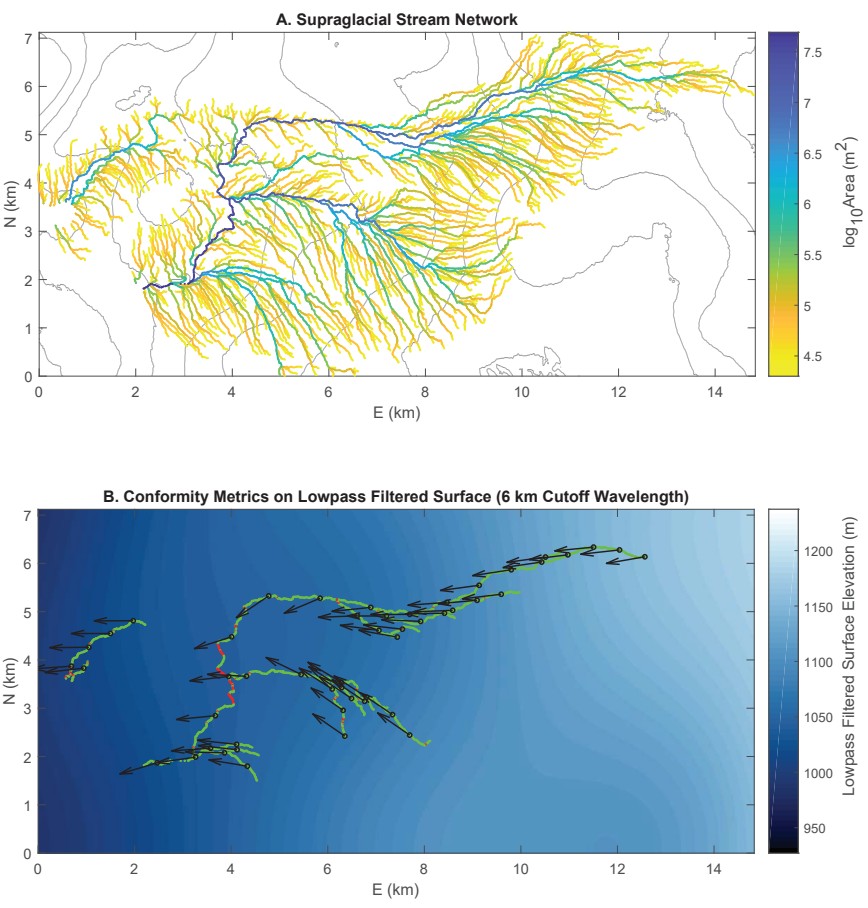

**Figure 6.** (A) Supraglacial stream network obtained by flow-routing on 2 m DEMs from study region R1 (Fig. 1.A, Table 1), colored by accumulated upstream drainage area. Surface elevation is shown with 20 m black contours. Fluvial incision rate should increase with increasing slope and drainage area (Eq. 7). (B) Illustration of stream conformity metrics for select streams from the same network, projected onto topography lowpass filtered at a 6 km cutoff wavelength. Sections of streams that would be flowing uphill on this filtered surface are colored red and other sections are green; this data is used to calculate percent downhill $\%d$. Black arrows indicate steepest descent directions on this filtered surface; the angle between these directions and the corresponding stream channel orientations is used to calculate conformity factor $\Lambda$.

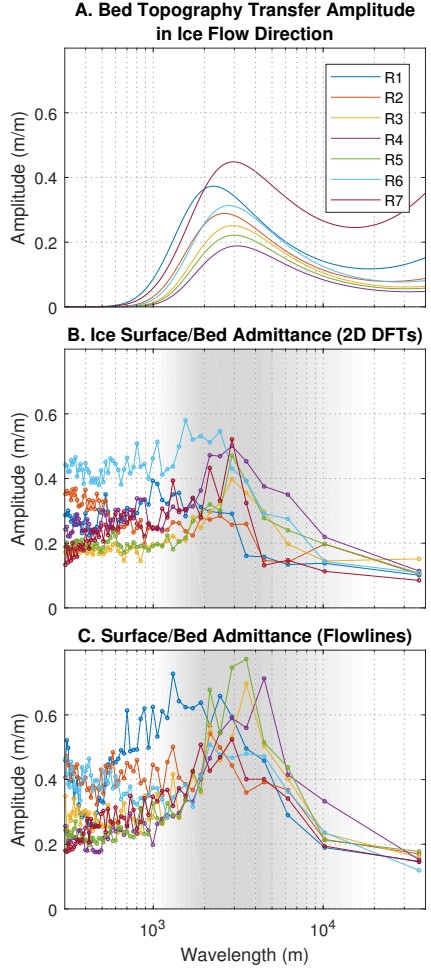

**Figure 7.** (A) Predicted bed topography transfer amplitudes from our seven study regions (Fig. 1.A, Table 1), with $\eta = 10^{14}$ Pa s and $C^{0*} = 10$ in all regions. (B) Results from one method for calculating empirical admittance of measured bed topography to observed ice surface topography by binning and averaging absolute wavenumber components of 2D DFTs (see Section 2.3.5). (C) Results from a second method for calculating empirical bed-to-surface admittance by of binning and averaging 1D DFTs from multiple ice flowline transects. For all regions both calculations of empirical admittance (B and C) generally match predicted transfer amplitudes (A) at wavelengths greater than $\sim 1$ km and exhibit similar peaks between $\sim$1-10 km (shaded regions in B and C). At wavelengths less than $\sim 1$ km both empirical admittance calculations are higher than predicted transfer amplitudes.

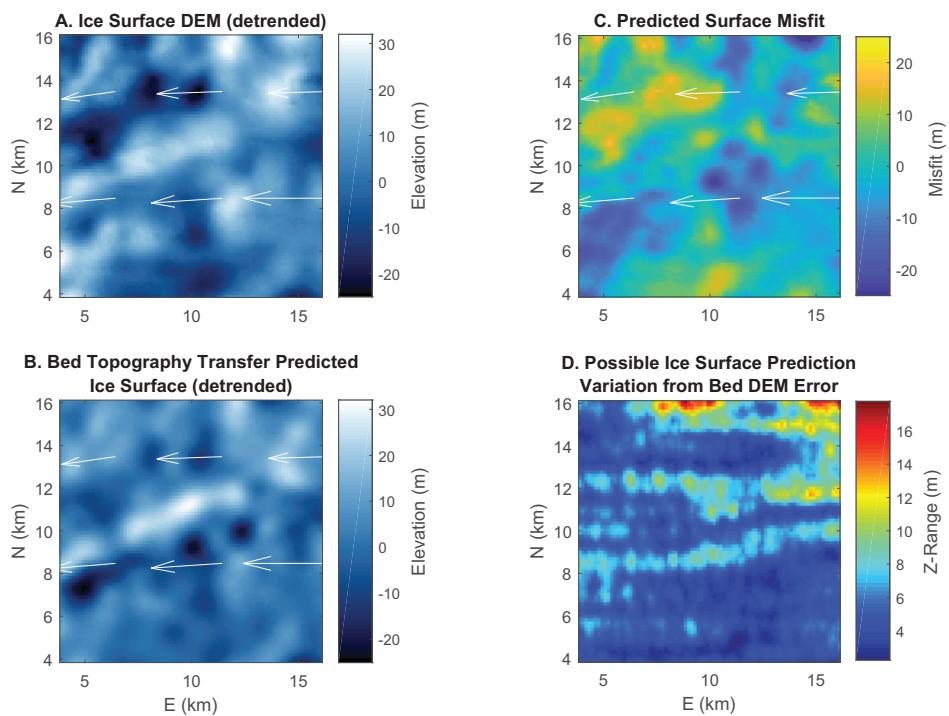

**Figure 8.** Example ice surface prediction and error analysis from study region R1 (Fig. 1.A, Table 1). (A) Detrended ice surface DEM. (B) Detrended bed topography transfer predicted ice surface, with $\eta = 10^{14}$ Pa s and $C^{0*} = 10$. We note that km-scale depressions and ridges/peaks are generally configured similarly to the real ice surface DEM in plot A, and that topographic relief also corresponds well. (C) Prediction misfit (subtraction between the actual and predicted ice surfaces in plots A and B). Prediction misfit is often significant (mean misfit in this region is 13.3% of the regional topographic relief), which might be expected for a number of reasons discussed in Sections 2.1, 2.3.1, and 2.3.2. (D) Potential effects of bed DEM error on ice surface predictions (see error map in Fig. 2.A). Where bed DEM error is less than ~60 m the potential surface prediction variation is much less than the amplitude of surface topography. White arrows in all plots indicate ice surface velocity field, and the bed DEM underlying this region is shown in Fig. 1.B.

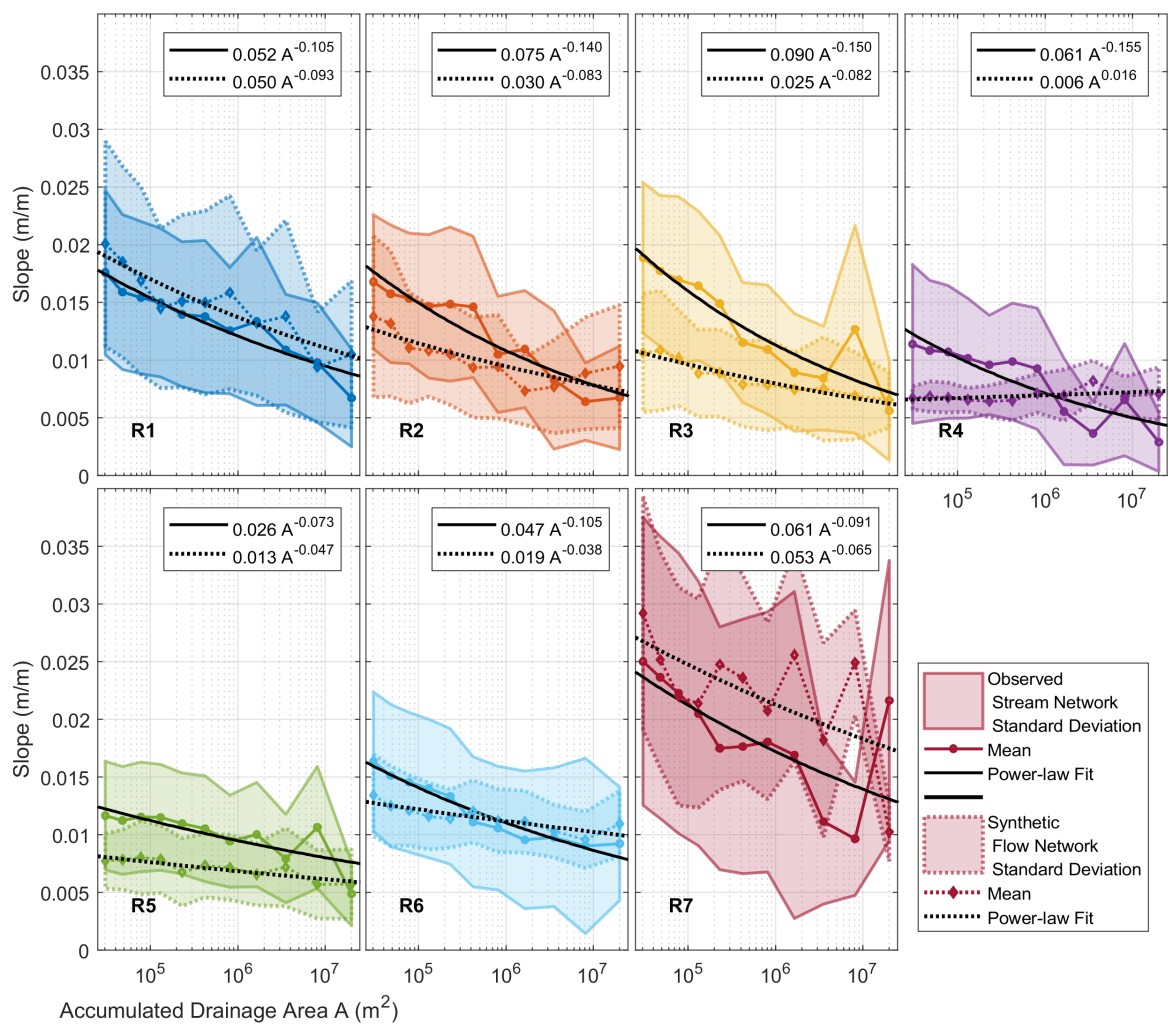

**Figure 9.** Mean stream channel slopes binned by accumulated upstream drainage/flow area from our seven study regions (Fig. 1.A, Table 1). Results are shown from both observed supraglacial stream networks and synthetic flow networks calculated on bed topography transfer predicted surfaces (see Sections 2.4.2 and 2.4.3). The standard deviation of slope within each area bin is indicated by shaded patches, where patches with solid outlines correspond to observed stream networks and patches with dotted outlines to synthetic flow networks. Mean slopes are shown by solid and dotted colored lines and power law fits by solid and dotted black lines. Under the (likely inaccurate) assumptions that supraglacial stream longitudinal elevation profiles are incisionally controlled and in a steady-state configuration under ice flow conditions analogous to uniform uplift, the power-law fit coefficients and exponents should correspond to $K^{-1}$ and $-m$ from equation 7. Synthetic flow networks from regions R3, R4, R5, and R6 may not be meaningful due to surface under-prediction (see Section 3.2).

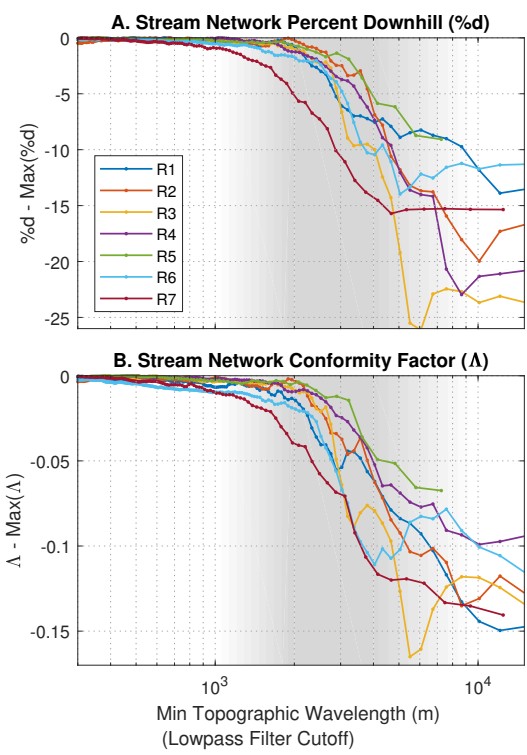

**Figure 10.** (A) Percent downhill $\%d$. (B) Conformity factor $\Lambda$. Values for both stream metric topographic conformity metrics are calculated in our seven study regions(Fig. 1.A, Table 1). All values are normalized to the maximum values in each network due to the variability in plateau values of $\%d$ and $\Lambda$ between networks. For all supraglacial stream networks the cutoff filter wavelengths over which $\%d$ and $\Lambda$ decrease most significantly are between $\sim$1-10 km (shaded regions), similar to the bed topography wavelengths predicted to transfer most strongly (7).

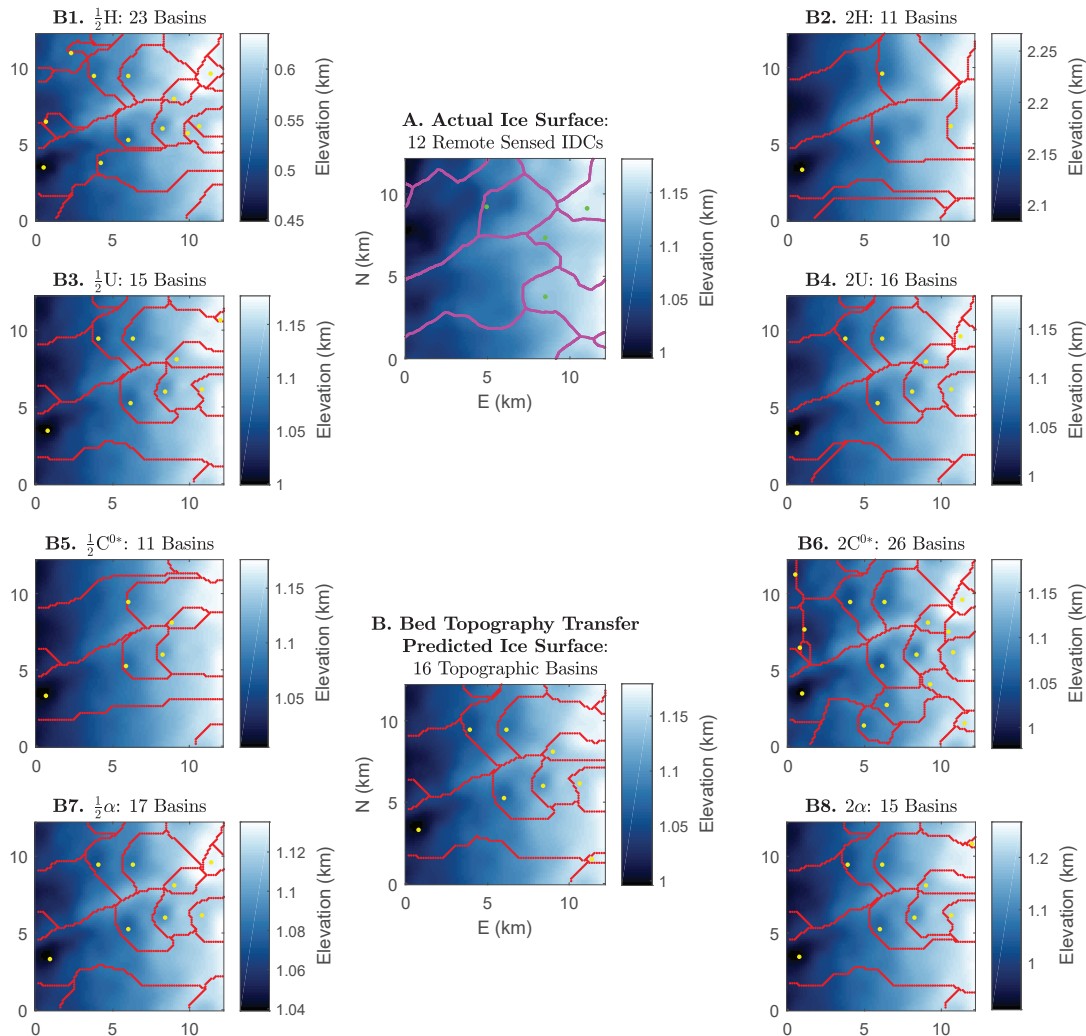

**Figure 11.** (A) IDCs (magenta outlines) and moulins (green dots) obtained from satellite images by (Yang and Smith, 2016). (B, B1-B8) Ice Surface topographic basins (red outlines) and local minima (yellow dots) on bed topography transfer predicted ice surfaces with various ice flow parameters. From study region R1 (Fig. 1.A, Table 1) with $\eta = 10^{14}$ Pa s and baseline $C^{0*} = 10$. While the bed transfer predicted topographic basin configuration in plot B is different from the IDC configuration in plot A, the basin densities are similar. Changing ice thickness $H$ or basal sliding parameter $C^{0*}$ by factors of two produces significant changes in predicted topographic basin configurations (B1-B2 and B5-B6).

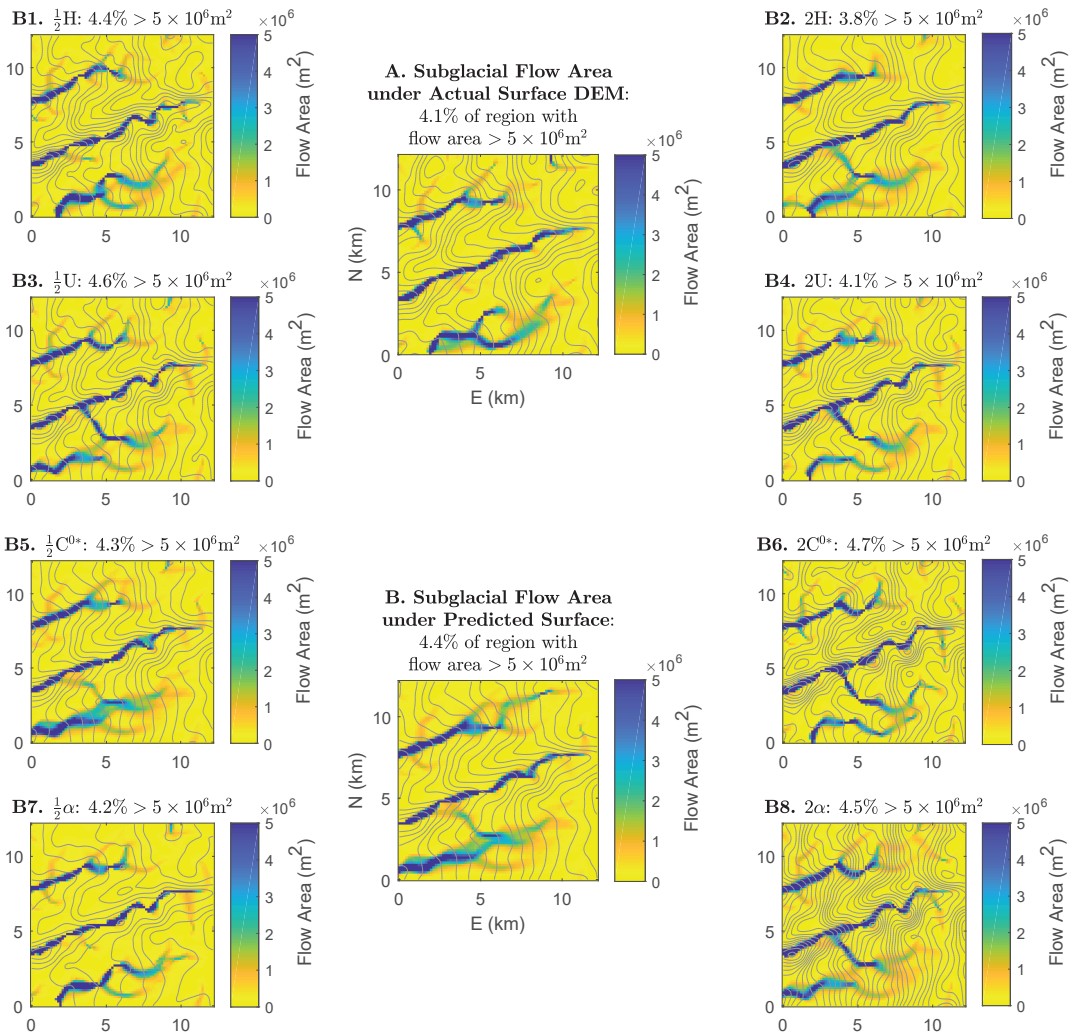

**Figure 12.** (A) Subglacial accumulated flow (drainage) area obtained via flow-routing on hydraulic potential fields calculated under the actual ice surface DEM from BedMachine/GIMP, (Howat et al., 2014; Morlighem et al., 2017a). Grey contours in all plots are 0.1 MPa hydraulic potential contours. (B, B1-B8) Subglacial accumulated flow area calculated under the bed topography transfer predicted ice surface with various ice flow parameters. From study region R1 (Fig. 1.A, Table 1) with $\eta = 10^{14}$ Pa s and baseline $C^{0*} = 10$. The flow-area threshold displayed ($5 \times 10^6$ m$^2$) was chosen to highlight pathways of high relative water flux.

**Table 1.** Study region information (locations in Fig 1.A) and bed topography transfer function surface prediction results. We note that the mean misfit metric used here is not a comprehensive indicator of fit quality, and that in regions R3, R4 and R5, and R6 surface topographic relief is significantly under-predicted. In all regions the transfer function surface predictions appear to be optimized with higher values of basal sliding than have been determined with other methods (MacGregor et al., 2016; Ryser et al., 2014b), we expect this is due to a combination of non-Newtonian rheology and poor bed DEM resolution.

| Study region | $H$ | $U$ | $\alpha$ | RACMO melt rate | Bed DEM mean error | Surface prediction mean misfit ($C^{0*} = 10$) | Best fitting $C^{0*}$ |
|---|---|---|---|---|---|---|---|
| | (m) | (m/yr) | (rad) | (mm water/yr) | (m) | (% topographic relief) | |
| R1 | 824 | 86 | 0.013 | 2107 | 28.9 | 13.3 | 10 |
| R2 | 1075 | 87 | 0.009 | 1145 | 30.0 | 9.35 | 6 |
| R3 | 1233 | 88 | 0.009 | 793 | 41.1 | 10.4 | 13 |
| R4 | 1335 | 81 | 0.006 | 556 | 80.0 | 12.4 | 29 |
| R5 | 840 | 214 | 0.017 | 1321 | 55.8 | 13.9 | 23 |
| R6 | 1266 | 87 | 0.007 | 711 | 53.2 | 11.2 | 11 |
| R7 | 1187 | 92 | 0.011 | 1552 | 49.1 | 13.7 | 11 |