# Peer review of "Basal control of supraglacial meltwater catchments on the Greenland Ice Sheet"

_The Cryosphere, 2018_

## Referee Comment (RC1) · Anonymous Referee #1 · 14 Jun 2018

General Comments

This paper uses previously established transfer functions to make three points:

1) that 1-10 km scale topography on the ice sheet is controlled by bed topography. I don't disagree with this statement because its more or less the conventional wisdom and others have demonstrated this to be the case. It is certainly not new and I don't feel the results presented really shed any new insight relative to Greenland.

2) Changes in sliding will radically alter the surface topography and catchments, leading to smaller catchments with more moulins and less efficient drainage. This point is somewhat of a stretch given that the high sliding really only occurs in the summer – most of the evolution of the glacier takes place over the other 9 or 10 months of the year.

[Figure]

Moreover, they appear to use a very high slip ratio of 11 given that from what I can tell its derived using winter velocities in regions with quite warm (perhaps even temperate ice), with high slopes, so one would expect deformation to be significant ($\sim$50/50 as Ryser et al, JGlac 2014 show). Ryser et al show slip ratios this high in summer, but only for a few brief peaks each summer (the annual average slip ratio is much lower). Citing this work as well as others on actual slip ratios would make sense. From Figure 6, its seems like the misfit is somewhat insensitive (broad minimum) to this parameter, so how is the ice sheet so sensitive to change in sliding. In short, the feedback they suggest between catchment size and sliding is not at all well supported. It's also not clear how much faith we should put in a theory derived for small perturbations applied to high-amplitude topography with a linear rheology in place of a non-linear rheology. Such cases can be illustrative, but one has to be careful about then inverting and assigning too much quantitative credence to the results.

3) There is a lot about thermal-erosion that's not really well explained. There numerous cases where major drainages are observed to be bridged due to large melt channels. So, I am not really sure what the major point is.

Nearly every Figure is referenced parenthetically, without ever explaining what the figure is supposed to be showing. Statements like "We computed xyz results to make some point. The results show that. . .." Would be helpful. The captions themselves are generally terse and don't really explain the figures well, especially without supporting explanation in the text. In some cases, the figures appear to be referred to out of order (5 before 4). With respect to the number of figures, this is probably a case of less is more (i.e., fewer, better explained, and more relevant figures).

The appendix seems to be largely a rehash of Gudmundson's work with a few symbols changed. A whole section to define Fourier transforms is unwarranted.

In summary, I don't see that this paper adds much new knowledge or insight in its present form. It probably needs a complete restructuring and rewrite.

Specific Points

P1/L18 – disperse -> dispersed

P1/L18/19 – more dispersed yes, but under the scenarios that would reach this point, the volume of melt water would be greater (i.e., warming world), so it is not clear whether the efficiency would increase or decrease.

P2/L18 – set however off with commas (, however,)

P2 L26/27 – would be appropriate to cite Joughin et al 2013 Cryosphere here (and perhaps elsewhere). Their paper has a quite a bit of discussion on the interaction of basal and surface topography and the effect of water routing.

P2 L31 – insert a comma before "which" P3 L6 – "it is unclear whether dynamic stream incision is efficient enough compared to other topographic influences to influence IDC-scale topography and meltwater routing" Not sure I understand this statement – as noted below, a quick google search can turn up many pictures see large stream channels cut by overtopping streams. P3 L24 – don't make Greenland Ice Sheet an acronym as GIS is to commonly used for mapping. You are not word constrained and in most cases you can be brief by just saying Greenland or the ice sheet. P2 paragraph that starts with L20 or L26 – there probably should be a reference to Smith, Raymond, and Scambos 2006, JGR F101019 as they look at the transfer of bed topography to the surface of the Greenland ice sheet. Their findings with respect to anisotropy would make sense to discuss later in the paper as well. P3 L31 – replace "resolution" with "posting" as you note in the next couple of sentences the resolution is anything but 150 m. Ditto for P4 L 2, and L3 (using sampling spacing if you want to avoid repetitive use of posting). P4 L4 – "All" to "The" P4 L14 – Define RSF. P5 16 – hyphenate no-flow condition P13 L26 add a "the" before "∼1-10" P13 L30 This is almost identically restates what was said 4 lines earlier. P14 L1-1 again somewhat repetitive and somewhat repeating the obvious that could be inferred from previous work with transfer functions and observations of bed and surface topography. P14 L16-17 – "If ice surface
Interactive
comment

adjustments to variable basal conditions or ice flow perturbations are sufficiently rapid, surface topographic basin configuration should also vary on seasonal timescales." If this were the case, then such changes should be occurring now. To the extent any such changes have occurred they escaped notice of numerous groups observing elevation time series. L14 L24-25 "basal sliding" its important to keep in mind the periods of strong basal sliding relatively brief and most of the year there is no surface melt, so this period of low sliding likely dominates the transfer of bed to surface topography. This statement also applies to the following paragraph. P15-L5-10 – again the winter pattern is likely to dominate and offsets any summer change with a wholesale redistribution of the drainage patterns. P15 Section 4.3 There is a significant amount of thermal-fluvial erosion – most stream channels are down-cut by by 10s of centimeters to meters. There are many examples of large stream channels – simple google meltwater stream channels Greenland and select the images tab. The really deep ones are not necessarily that common, but they often occur in locations where a major drainage catchment feeds a lake, that overtops, cut a channel many meters deep, to connect up with another drainage or to find a moulin. I think part of the problem with this section is that its poorly written and its not really clear the point the authors are trying to make.

---

## Referee Comment (RC2) · Anonymous Referee #2 · 5 Jul 2018

**Summary**

This paper explores the factors controlling the catchments of surface rivers on the western Greenland Ice Sheet. It focuses on the relationship between basal topography and these rivers, and concludes that certain geometric aspects (basal bumps and the basal slip ratio) control the organization of surface hydrology. From there, a a possible ice-flow feedback is hypothesized based on future projected changes in melt rate and slip ratio. The sign of the ice-flow feedback is unknown.

The study emphasizes the methods (Laplace-domain transfer functions) and most of the results presented (transfer function amplitudes) are a step away from reality, limiting the extent to which results are compared to data. Accordingly, the Results section is very brief (2 pages) compared to the rest of the manuscript (17 pages + Appendix) and

the Methods section (8 pages). Phrases like "as expected" of "consistent with previous work" appear frequently, highlighting that this study is light on novel contributions. Most of the 3-page Discussion is speculative and only loosely constrained by the results presented.

The study design is flawed in that the root data (Morlighem bed DEM) are **not independent** of the validation data (ArcticDEM for the ice-sheet surface) in the regions the authors chose to study (which are, incidentally, areas where Morlighem applied mass conservation). The authors also studied one area (R7) where mass conservation was not applied; results there are not shown, but I would expect their predicted surface to more poorly match the true (Arctic DEM) surface. This is hinted at in Figure 9, but never addressed. The techniques used to generate the bed DEM must be considered in this analysis; preferably, multiple regions with bed DEM constructed from mass conservation and with kriging should be analyzed and compared to one another.

The primary question of the study (see first sentence of this review) is interesting and potentially compelling over the next hundred years or so. However, the methods address it incompletely (surface processes, such as fluvial erosion, are only speculated on) and suffer from a considerable flaw in using a bed DEM informed by the surface topography. The paper is out of balance and difficult to follow. If the authors can restructure the manuscript, address the data issues, and either refocus the central question on basal control alone or add treatment of surface-based topographic controls, this would become a worthy contribution.

**Specific comments**

The best-fit value of $C^{0*}$ = 11 reported here is, as pointed out by Reviewer 1, anomolously high compared to field observations. This is especially important because the authors identify $C^{0*}$ as a parameter that IDC density is most sensitive to (Figure 12); thus, it would seem crucial to use a realistic value of $C^{0*}$. The authors' finding of $C^{0*}$ = 11 by their techniques thus suggests either (1) that other techniques

should be used to find a more realistic $C^{0*}$ before this analysis is continued, or (2) if the authors are confident in $C^{0*}$ = 11, the meaning and implications should be explored, which could be an interesting result.

A good paper can demonstrate much of its message through its figures alone. In this case, it is hard to follow the meaning of the figures, which are too many in number (12) and too focused on the methods, which are already well established (Gudmundsson 2003 and other work since). However, this can be readily improved. Recommendations for the figures:

Figure 1: Adapt but keep. Zoom in better on Panel A. Add labels to Panel B – is this the ArcticDEM surface, or a predicted surface?

Figure 2: Unnecessary and repetitive from earlier work, remove. Phase is not a big part of the analysis, consider discarding or at least deemphasizing (no figures on phase).

Figure 3: Could adapt and keep. Is panel B correct: thin ice (H=500 m) will best express bedrock features of 100 km scale?

Figure 4: Unnecessary; remove.

Figure 5: Unnecessary; remove all except Panel E, which could be incorporated into Figure 3.

Figure 6: Potentially useful, but why is the misfit pattern so sensitive to $\eta$? How many values of $\eta$ were tested, and why is the misfit so concentrated at $10^{15}$ Pa s? Not what I would expect.

Figure 7: Keep; make color scales the same on Panels E and F.

Figure 8: Panel B is not useful. Are the data shown in Panel A from this paper, or previous work? Consider deleting entirely.

Figure 9: Panel A is misleading because the Stream Free Region (RSF) looks different from all other regions, which may be intended to show the influence of streams. Yet

the cause is simply much different H, u, and $\alpha$ in this region compared to other regions (Table 1). For better fidelity, the authors should select a RSF with similar ice geometry to the stream regions.

Figure 10: Even after a lot of thought, I am still not sure what is being plotted here. I understand the meanings of $\%d$ and $\Lambda$, but cannot understand the choice on the y-axis (difference from maximum). In the text, a normalized framework (0 to 1) is discussed, but the data here are not shown that way. The text also highlights variability at small wavelengths (P13 L6-11), but the figure presentation makes this information uninterpretable (all curves are plotted too densely at low wavelengths). Regardless, I infer that the point of this figure is to show the natural variability in both $\%d$ and $\Lambda$, by showing the values across R1-R7, and then comparing to the flow networks. For $\Lambda$, the flow networks fall within the natural variability, but for $\%d$, it does not. This could suggest something about the control of fluvial erosion, or other surface processes, on surface topography, but this is not addressed.

Figure 11: Panel A is not necessary, but Panel B presents a comparison of inferred surface to actual surface, which is essential. Why were such comparisons not run on all 7 study regions? Yet, the text (P13 L16-20) declares the slope-area metric to be of limited utility, according to previous work and data from this study. Thus, any conclusions based on this data (P13 L21-27, P14 L1-2) should be de-emphasized or removed.

Figure 12: Keep.

I also suggest better distinguishing what is observationally based (e.g., stream networks) from what is computed here using transfer functions (e.g., flow networks).

The main conclusion, that "bed topography transfer alone can explain ∼1-10 km scale ice sheet surface topography" (P13 L29-30), is not illustrated well in any one figure. It can perhaps be inferred from Figure 7E, but the spatial scale must be eyeballed, rather than shown as the independent variable like in the majority of the figures.

The Discussion section is largely uncoupled from the rest of the work. Speculation on changes in basal slipperiness on hourly to seasonal timescales (P14 L28, L33, P15 L1-2) is not relevant to the bed-to-surface propagation this paper addresses, as the stated timescale for adjustment is >3 years (P14 L14). Thus, the hypothesized feedback (increasing melt changes sliding, which changes IDC size, affects local melt water volumes at the bed, which again changes sliding), which operates on seasonal or shorter timescales, is not supported or constrained by the study. It would be an interesting concept if it could be shown, but that is not accomplished here.

The first paragraph of Section 4.1 reads like the main motivation for the study, and as such should appear in the Introduction.

Equations 7 and 8 appear in the Discussion, which is strange, and are not applied to further analysis. They should be removed.

The ideas on fluvial erosion (P16 L1-30) are potentially interesting, but again, are completely unexplored in the work. The statement "Our conformity metric calculations (Fig. 10) are consistent with an external control on supraglacial stream network geometry" (P16 L20-21) is not supported by the work. It may or may not be true, but the data were not shown to demonstrate it.

Overall, the base idea is worthy of exploration, but the paper is light on results and heavy on unsupported and speculative discussion, and does not fully consider the limitations of its primary dataset.

---

## Author Comment (AC1) · 22 Jul 2018

We would like to thank both referees for their constructive comments on our submission. We here present point-by-point responses to their comments. We expect that implementing the changes we outline in these responses will produce a greatly improved manuscript that informs bed-to-surface connections on the Greenland Ice Sheet, and supraglacial hydrology generally. We will better highlight the contributions of our study:

(1)      A direct verification that bed topography is the primary control on IDC-scale Greenland Ice Sheet surface topography using newly available bedrock DEMs, surface DEMs, and velocity data.

(2)      Demonstration with several geomorphological metrics that the general IDC-scale structure of supraglacial stream networks and drainage basins can also be explained by bed topography

(3)      Predictions of supraglacial drainage configurations/IDC densities that would occur in different long-term averaged ice flow conditions, illustrating that supraglacial drainage configurations can change significantly with changing ice flow conditions. This could have implications for ice sheet evolution, particularly if subglacial hydrology influences ice flow conditions.

Key changes in our manuscript will include restructuring to better contextualize the geomorphology/meltwater routing parts of our analysis, more thoroughly explaining our model, data, and parameter choices (including addressing seasonality, bed DEM limitations, and slip ratio values), more clearly highlighting our results, and making our figures and discussions more concise.

We note that in these responses all references to specific lines/sections of our manuscript refer to the original submission.

**Response to Referee #1:**

*This paper uses previously established transfer functions to make three points:*

*1) that 1-10 km scale topography on the ice sheet is controlled by bed topography. I don't disagree with this statement because its more or less the conventional wisdom and others have demonstrated this to be the case. It is certainly not new and I don't feel the results presented really shed any new insight relative to Greenland.*

That bed topography controls or strongly influences ice surface topography has been mentioned previously in literature. We have cited many of the authors that discuss this link and will add any that we missed. However, we disagree that our work is not useful in the context of Greenland. We are not aware of any publications explicitly and quantitatively testing this idea using 2D bedrock DEMs, or of any publications quantitatively examining how well bed topography explains 2D ice sheet surface topography as a function of wavelength. Our approach is useful in that we have demonstrated that a relatively simple and easy-to-implement analytical model can explain most IDC-scale (1-10 km) surface topography. The methodology we use is imperfect for reasons discussed in the manuscript (see sections 2.1, 2.3.1, and 2.3.2 in the original submission) and pointed out by the reviewers; we will cover some of these limitations more quantitatively in a revision and suggest ways to improve the analysis in future work.

*2) Changes in sliding will radically alter the surface topography and catchments, leading to smaller catchments with more moulins and less efficient drainage. This point is somewhat of a*

*stretch given that the high sliding really only occurs in the summer – most of the evolution of the glacier takes place over the other 9 or 10 months of the year.*

This hypothesis - that changes in surface drainage basins could affect subglacial drainage efficiency which would then feedback to surface topography - is supported by the sensitivity of bed transfer functions to changes in basal conditions through time. Testing this hypothesis would be an interesting direction for future work. The seasonal dynamics of ice sheets are of course incredibly important, but we do not attempt to model such short time scales in this work. Seasonal dynamics may be part of the reason why the ice surface topography we predict with transfer functions often exhibits not insignificant deviation from surface DEMs, and we will acknowledge this in a revision.

However, the transfer functions indicate that the wavelengths of features we focus on should change over timescales of 3-60 years, with minimal seasonal variation (section 2.3.2). We also point out (consistent with the referee's later statement) that we are not aware of observations indicating that IDC-Scale ice surface topography generally changes significantly on a seasonal basis. If the ice surface topography doesn't change significantly on a seasonal basis but other ice flow parameters do vary, then it is reasonable to question how the "long term, effective," ice flow parameters that govern surface topography relate to the dynamic ice flow parameters (i.e., annual average or peak values). We do not attempt to address this in our work.

*Moreover, they appear to use a very high slip ratio of 11 given that from what I can tell its derived using winter velocities in regions with quite warm (perhaps even temperate ice), with high slopes, so one would expect deformation to be significant (~50/50 as Ryser et al, JGlac 2014 show). Ryser et al show slip ratios this high in summer, but only for a few brief peaks each summer (the annual average slip ratio is much lower). Citing this work as well as others on actual slip ratios would make sense. From Figure 6, its seems like the misfit is somewhat insensitive (broad minimum) to this parameter, so how is the ice sheet so sensitive to change in sliding. In short, the feedback they suggest between catchment size and sliding is not at all well supported. It's also not clear how much faith we should put in a theory derived for small perturbations applied to high-amplitude topography with a linear rheology in place of a non-linear rheology. Such cases can be illustrative, but one has to be careful about then inverting and assigning too much quantitative credence to the results.*

A slip ratio of 11 is indeed at the high end of the ranges presented for our study regions in other publications (such as the suggested Ryser et al 2014 or our original reference MacGregor et al 2016). We chose a slip ratio of 11 because 11 was the mean best fitting value found from inversions in all study regions with reliable inversion results. However, the inversion minima is often broad (as indicated in figure 6), and so in some regions we could choose a slip ratio as small as around 4 without obtaining significantly worse surface predictions. We will present more inversion results in a revision to illustrate this.

We expect that the high values of slip ratio we found primarily reflect the assumption of Newtonian ice rheology that is used in the transfer functions, rather than the true basal slip. The analysis of Raymond and Gudmundsson (2005) shows that non-linear rheologies generally increase transfer amplitude peak for a given sliding value, thus linear rheology will predict larger basal slip ratios to attain a given transfer peak amplitude. Raymond and Gudmundsson (2005) also demonstrate that the shape of the transfer function as a function of wavelength is quite similar between Newtonian and power law ice

rheology. As we show in section 4.1 and figure 12, changing slip ratio does alter surface topographic basin configuration. The importance of slip ratio is not inherently inconsistent with the broad constraints that our inversions place on slip ratio, which reflects both bedrock DEM errors and simplifying model assumptions. Because of this, we focus on relative changes of basal transfer with changing ice flow parameters. By showing that different regions of the GIS with a range of ice flow parameters are well modeled, we can extrapolate to predict approximate changes in surface topography (and basal hydraulic potential) upon varying ice flow parameters. In other words, we quantify how well the transfer functions work, show that they produce surface topography with sufficient accuracy as to be useful for making general predictions, and use them as a means to quantitatively examine ideas about meltwater routing.

Our study thus overpredicts basal slip ratio, but reasonably accurately predicts how variations in basal slip modify surface topography. Indeed, using smaller values for slip ratio would not greatly impact some of our primary conclusions so long at the slip ratios used are not smaller than around 4. For the amplitude spectra comparisons we covered in sections 2.3.5, 3.1, and figure 9, using a smaller slip ratio does not strongly change the general shape of the transfer function or location of the transfer amplitude peak. We have included a modified version of figure 9 in our manuscript (Response Figure 1) to show that for ice flow parameters representative of our region of interest we still predict similar 1-10 km transfer peaks with a slip ratio of 4. Since the wavelengths of peak bed transfer are relatively insensitive to sliding, our conclusion that bed topography transfer can explain the conformity metrics would still be valid (section 3.2.1 and figure 10). Additionally, while the slope of our calculated slope-area trends on synthetic flow networks (figure 11) might change slightly if we predict surfaces using lower slip ratios, the main point that bed topography transfer alone can create negative slope-area relations like those observed in stream networks will still be valid (section 3.2.2). Our goal is not to match the exact slope-area trends of supraglacial networks here, although such work (which might involve modeling ice flow in greater detail and coupling this with surface fluvial incision) will be a rich area for future study. We will comment on this in the discussion.

[Figure]

**Response Figure 1.** Demonstration that lower values of slip ratio ($C^{0*}$) still predict bed topography transfer peaks at wavelengths from 1-10 km (panel B), but less effectively match observed admittances. This figure is similar to figure 9 in our manuscript except for panel B.

*3) There is a lot about thermal-erosion that's not really well explained. There numerous cases where major drainages are observed to be bridged due to large melt channels. So, I am not really sure what the major point is.*

We will remove extraneous discussions of thermal-erosion, and restructure the rest to better illustrate the relation to bed topography transfer and to our key points. We are not claiming that large-scale surface topographic features fully control stream network and drainage basin structure, and the possibility that fluvial processes contribute to internally drained basin reorganization on a seasonal scale is not ruled out by our work. Rather, we find that, in all the regions we examined, the general spatial density, size, and approximate configuration of drainage basins is well explained with surface topographic wavelengths > 1 km, which is the range of length-scales over which bedrock transfer is effective. We did not attempt to carry our more rigorous statistical studies over larger regions, but this will become more tractable as bed DEM data improves. The fact that we can predict the large-scale basin structure reasonably well from only bed topography, even using a fairly simple ice flow model,

verifies that basal processes are the first-order control on IDC-scale surface topography and meltwater routing, a point that has important implications which we explore in our discussion.

> *Nearly every Figure is referenced parenthetically, without ever explaining what the figure is supposed to be showing. Statements like "We computed xyz results to make some point. The results show that. . .." Would be helpful. The captions themselves are generally terse and don't really explain the figures well, especially without supporting explanation in the text. In some cases, the figures appear to be referred to out of order (5 before 4). With respect to the number of figures, this is probably a case of less is more (i.e., fewer, better explained, and more relevant figures).*

We will better contextualize figure references and add more explanation to the captions. As readers can reference Gudmundsson 2003 for similar figures similar to 2 and 4 illustrating the transfer functions, we will place these in a supplement. As figure 5 illustrates only an accessory point about our methods, we will consider also removing this figure. We will add a figure to better illustrate the stream conformity metrics.

> *The appendix seems to be largely a rehash of Gudmundson's work with a few symbols changed. A whole section to define Fourier transforms is unwarranted.*

We will remove or consolidate these appendices.

> *In summary, I don't see that this paper adds much new knowledge or insight in its present form. It probably needs a complete restructuring and rewrite.*

We agree that some restructuring will be beneficial in making our points more clear and better supporting them. Though there is of course much room for future improvements, we do believe that our work makes three primary and worthwhile contributions, as stated in the introduction to these responses.

> *Specific Points*
>
> *P1/L18 – disperse -> dispersed P1/L18/19 – more dispersed yes, but under the scenarios that would reach this point, the volume of melt water would be greater (i.e., warming world), so it is not clear whether the efficiency would increase or decrease.*

A good point, we will add this to section 4.2 of our discussion. We will also more explicitly point out that while an examination of potential subglacial-supraglacial feedbacks is a natural extension of our work, there are basic aspects of subglacial hydrology that are still poorly known. Our approach simply indicates that there is plausibly some feedback between surface topography, surface hydrology, and basal hydrology.

> *P2/L18 – set however off with commas (, however,)*
>
> *P2 L26/27 – would be appropriate to cite Joughin et al 2013 Cryosphere here (and perhaps elsewhere). Their paper has a quite a bit of discussion on the interaction of basal and surface topography and the effect of water routing.*

Will implement both of the above suggestions.

*P2 L31 – insert a comma before "which" P3 L6 – "it is unclear whether dynamic stream incision is efficient enough compared to other topographic influences to influence IDC scale topography and meltwater routing" Not sure I understand this statement – as noted below, a quick google search can turn up many pictures see large stream channels cut by overtopping streams.*

Will clarify the statement, the intent was to indicate that it has not yet been demonstrated how significant of a role thermal-fluvial incision plays in setting the large-scale structure and evolution of subglacial drainage basins, relative to how much of this structure is primarily set by bed topography. Certainly there are local examples of fluvial erosion influencing drainage patterns, but the effect of bed topography filtered through to the surface appears to be the primary influence on larger scales, and is generally of larger amplitude than seasonally averaged fluvial incision.

*P3 L24 – don't make Greenland Ice Sheet an acronym as GIS is to commonly used for mapping. You are not word constrained and in most cases you can be brief by just saying Greenland or the ice sheet. P2 paragraph that starts with L20 or L26 – there probably should be a reference to Smith, Raymond, and Scambos 2006, JGR F101019 as they look at the transfer of bed topography to the surface of the Greenland ice sheet. Their findings with respect to anisotropy would make sense to discuss later in the paper as well. P3 L31 – replace "resolution" with "posting" as you note in the next couple of sentences the resolution is anything but 150 m. Ditto for P4 L 2, and L3 (using sampling spacing if you want to avoid repetitive use of posting). P4 L4 – "All" to "The" P4 L14 – Define RSF. P5 16 – hyphenate no-flow condition P13 L26 add a "the" before "~1-10" P13 L30 This is almost identically restates what was said 4 lines earlier. P14 L1-1 again somewhat repetitive and somewhat repeating the obvious that could be inferred from previous work with transfer functions and observations of bed and surface topography.*

Will make changes according to the above suggestions.

*P14 L16-17 – "If ice surface adjustments to variable basal conditions or ice flow perturbations are sufficiently rapid, surface topographic basin configuration should also vary on seasonal timescales." If this were the case, then such changes should be occurring now. To the extent any such changes have occurred they escaped notice of numerous groups observing elevation time series.*

We included this statement to indicate that our "long-term averaged" predictions could miss seasonal dynamics; if we leave this statement in the discussion we will point out that such an effect has not been observed, and that this is consistent with the timescales predicted by the transfer functions. We are not aware of any studies that have explicitly examined this (point GPS measurements are not ideal for basin-scale deformation), but if there are generally no significant changes in supraglacial topographic basin configuration then our long-term averaged surface predictions could be considered more robust, since there would be a lower potential for inaccuracy due to seasonal dynamics.

*L14 L24-25 "basal sliding" its important to keep in mind the periods of strong basal sliding relatively brief and most of the year there is no surface melt, so this period of low sliding likely dominates the transfer of bed to surface topography. This statement also applies to the following paragraph. P15-L5-10 – again the winter pattern is likely to dominate and offsets any summer change with a wholesale redistribution of the drainage patterns.*

We will clarify in both locations that we are referring to changes to "average effective basal sliding". We cannot say from our current methods/data if just winter slip ratio influences topography, or if seasonal speedups matter. However, it does not seems unreasonable to propose that summer sliding rates might have some impact on the long-term averaged ice sheet surface topography, unless the ice surface fully readjusts to changing flow each season (which as previously discussed seems unlikely). We additionally note that long-term changes in atmospheric temperatures could change the length of time each year over which sliding occurs, whether or not sliding rates are affected.

> *P15 Section 4.3 There is a significant amount of thermal-fluvial erosion – most stream channels are down-cut by by 10s of centimeters to meters. There are many examples of large stream channels – simple google meltwater stream channels Greenland and select the images tab. The really deep ones are not necessarily that common, but they often occur in locations where a major drainage catchment feeds a lake, that overtops, cut a channel many meters deep, to connect up with another drainage or to find a moulin. I think part of the problem with this section is that its poorly written and its not really clear the point the authors are trying to make.*

We have indeed published on the potential influence of fluvial erosion on Greenland ice sheet topography (Karlstrom et al., JGR, 2013, Karlstrom and Yang, GRL, 2016). This influence is undeniable on small scales as you mention. But bed topography filtered through to the surface is of larger amplitude than seasonal melt in most places, so basin-scale structures are essentially static year to year - this may also be seen clearly on Google Earth (also see Fig 1 of Karlstrom and Yang 2016). We are focusing on what controls basin-scale surface hydrology here. We will restructure the text to make this more clear.

**Response to Referee #2:**

*Summary*

*This paper explores the factors controlling the catchments of surface rivers on the western Greenland Ice Sheet. It focuses on the relationship between basal topography and these rivers, and concludes that certain geometric aspects (basal bumps and the basal slip ratio) control the organization of surface hydrology. From there, a possible ice-flow feedback is hypothesized based on future projected changes in melt rate and slip ratio. The sign of the ice-flow feedback is unknown.*

*The study emphasizes the methods (Laplace-domain transfer functions) and most of the results presented (transfer function amplitudes) are a step away from reality, limiting the extent to which results are compared to data. Accordingly, the Results section is very brief (2 pages) compared to the rest of the manuscript (17 pages + Appendix) and the Methods section (8 pages). Phrases like "as expected" of "consistent with previous work" appear frequently, highlighting that this study is light on novel contributions. Most of the 3-page Discussion is speculative and only loosely constrained by the results presented.*

We attempted to use the best data-sets available, but given that there were still limitations in data quality for important factors such as bed elevation, we chose modeling approaches (basal transfer functions and surface flow routing) that are simple to understand, apply, and generalize despite limitations in accuracy. Our evaluation and presentation metrics (amplitude spectra, slope-vs-drainage area trends, and stream network conformity values) capture general traits of surface topography and stream networks that are robust in data and should not depend greatly on our model simplifications. These metics form the basis for a quantitative verification, using new datasets over multiple regions of the Greenland Ice Sheet ablation zone, of the extent to which bed topography explains surface topography and surface hydrology. Such a verification sets the stage for future more fully mechanistic studies. Our approach also permits the (testable) prediction of surface topography and drainage basin configurations in different ice flow conditions. These predictions indicate that changing ice flow conditions can appreciably affect supraglacial IDC configurations, which is a novel and significant point that we hope will spark further study. We agree that a more thorough presentation of results and a restructuring of discussions is warranted.

*The study design is flawed in that the root data (Morlighem bed DEM) are not independent of the validation data (ArcticDEM for the ice-sheet surface) in the regions the authors chose to study (which are, incidentally, areas where Morlighem applied mass conservation). The authors also studied one area (R7) where mass conservation was not applied; results there are not shown, but I would expect their predicted surface to more poorly match the true (Arctic DEM) surface. This is hinted at in Figure 9, but never addressed. The techniques used to generate the bed DEM must be considered in this analysis; preferably, multiple regions with bed DEM constructed from mass conservation and with kriging should be analyzed and compared to one another.*

The primary regions we examine are areas where mass conservation (based on surface elevation, velocity, and surface+bed mass balances) is used in conjunction with radar data to derive BedMachine DEMs. As discussed in section 2.2, we are interested in regions that exhibit significant supraglacial stream network development (typically at moderate elevations), that have accurate bed

elevation data, near-uniform velocities, and high resolution surface DEMs. Such regions of the ice sheet seem to generally be where mass conservation instead of kriging was used. However, we expect that BedMachine is the best choice for our study due to three reasons:

(1)      As far as we are aware, BedMachine is the most accurate Greenland bed DEM currently available, due in part to its use of mass conservation modeling which has advantages described in Morlighem et al (2011) and Morlighem et al (2014).

(2) The method used in derivation of the BedMachine DEM only considers mass conservation, and is thus fundamentally different from the Gudmundsson transfer functions which are derived from both mass conservation and the Stokes flow equations.

        We provide a thought experiment to demonstrate the distinction: given a bed elevation DEM, we could apply the BedMachine mass conservation method in reverse to predict steady-state surface elevations. Doing this uniquely would require a full surface velocity map, in addition to target values of background thickness and surface slope (and basal/surface mass balances if those are not assumed to be zero). The transfer functions make surface elevation predictions given the same target thickness and slope values, but just a single background surface velocity vector and a value/values for slip ratio. The transfer functions are thus also independently solving for a full velocity field by incorporating approximations of how ice should flow in response to gravity and pressure gradients. If we fed the same data we use to implement the transfer functions into an "inverse BedMachine model", by using single velocity vector and uniform mass balances over the whole domain, the methods would in general not predict the same surfaces.

        By verifying that the transfer functions can reasonably well predict the ice surfaces in our study regions, what we are verifying is that the approximations used to derive of the transfer functions are reasonably effective at least over 1-10 km scales. Furthermore, since where mass conservation modeling was used in the BedMachine DEM the mass balance terms were not perturbed to account for spatially nonuniform processes like fluvial incision, we are verifying that most of the ice surface topography at these scales is consistent with ice flow alone.

(3)      We focused on regions with relatively dense radar coverage, so the influence on mass conservation derived DEMs of surface data will be limited. The analysis we use for testing the possible effect of bed DEM error on our surface predictions, covered in section 2.3.4, 3.1, and figure 7 panels B and F, provides an indirect indicator of how sensitive our surface predictions are to the mass conservation modeling used in the BedMachine DEMs. This is the case because the published BedMachine error generally increases with increasing distance from radar data points. See the included response figure 2 for further elaboration on this. Our error analysis thus essentially demonstrates that in regions of dense radar coverage, the large-scale surface depressions/ridges we focus on should not be too significantly influenced by the topography between radar transects, and thus by the extrapolation method is used.

        We could attempt to obtain raw radar data for our study areas and interpolate it into DEMs using a method like kriging, but this would mean using less accurate bed elevation values (Morlighem et al 2014), and it is not clear that doing so would significantly impact our results.

[Figure]

**Response Figure 2.** Left panel: Error in the BedMachine v3 bed DEM generally increases with increasing distance from radar data (all bed picks from CReSIS shown in black), and our study regions (magenta boxes) are in areas where error is mostly less than around 100 m.

Center panel: Bedmachine elevations are generally in agreement with CReSIS radar data (radar data is colored by interpolated distance from BedMachine values). Some of the deviation of radar elevation points from BedMachine values may be due to error in the radar picks. Radar picks separated by hundreds of meters or less often exhibit hundreds of meters in elevation difference, meaning that a DEM perfectly conforming to radar data would frequently exhibit extremely (likely artificially) high relief.

Right panel: Lindback et al (2014) produced a Greenland bed DEM without mass conservation modeling. In places this differs appreciably from BedMachine, but where the DEM overlaps with our study regions they mostly do not differ too significantly (generally < 100 m). We note that BedMachine v3 incorporates more recent radar data than was available for the Lindback et al DEM.

Overall, this figure demonstrates that the BedMachine DEM agrees reasonably well with radar data, and that using an alternatively derived DEM would not necessarily benefit our study. We will consider adding a similar figure to a revision to show the important limitations of currently available bed DEMs.

*The primary question of the study (see first sentence of this review) is interesting and potentially compelling over the next hundred years or so. However, the methods address it incompletely*

*(surface processes, such as fluvial erosion, are only speculated on) and suffer from a considerable flaw in using a bed DEM informed by the surface topography. The paper is out of balance and difficult to follow. If the authors can restructure the manuscript, address the data issues, and either refocus the central question on basal control alone or add treatment of surface-based topographic controls, this would become a worthy contribution.*

We hope that our response to this review will provide adequate preliminary addresses to questions of data and methodology. We believe that significant restructuring, following what we outline in these responses, will highlight that our work does answer the question in the first sentence of this review, at least over the IDC-scales we focus on. Namely, we use multiple geomorphological metrics in combination with a simple method for predicting surface topography over an underlying bed to demonstrate that bed topography can generally explain surface catchment structure at IDC-scales, and that fluvial incision is thus a secondary and generally comparatively minimal influence at such scales. Using a more complex model for surface processes (such as fluvial incision) coupled with ice flow would be a natural and potentially interesting extension of our work, but we expect the results and ideas we present here are still novel contributions and are useful in part because of their generality.

*Specific comments*

*The best-fit value of $C_{0*} = 11$ reported here is, as pointed out by Reviewer 1, anomolously high compared to field observations. This is especially important because the authors identify $C_{0*}$ as a parameter that IDC density is most sensitive to (Figure 12); thus, it would seem crucial to use a realistic value of $C_{0*}$. The authors' finding of $C_{0*} = 11$ by their techniques thus suggests either (1) that other techniques should be used to find a more realistic $C_{0*}$ before this analysis is continued, or (2) if the authors are confident in $C_{0*} = 11$, the meaning and implications should be explored, which could be an interesting result.*

Refer to our earlier response to Referee 1.

*A good paper can demonstrate much of its message through its figures alone. In this case, it is hard to follow the meaning of the figures, which are too many in number (12) and too focused on the methods, which are already well established (Gudmundsson 2003 and other work since). However, this can be readily improved. Recommendations for the figures:*

*Figure 1: Adapt but keep. Zoom in better on Panel A. Add labels to Panel B – is this the ArcticDEM surface, or a predicted surface?*

We will implement these suggestions (the ice surface in figure B is from ArcticDEM).

*Figure 2: Unnecessary and repetitive from earlier work, remove. Phase is not a big part of the analysis, consider discarding or at least deemphasizing (no figures on phase).*

We agree that this figure is not particularly necessary, and will remove it.

*Figure 3: Could adapt and keep. Is panel B correct: thin ice (H=500 m) will best express bedrock features of 100 km scale?*

For all sets of ice flow parameters (excluding some values as zeros/infinities) as wavelength increases predicted transfer amplitude eventually approaches 1. However, we note that the transfer functions are not valid out to arbitrarily long wavelengths; at length scales very large compared to ice

thickness the background gravity current profile of flowing ice will dominate. We also note that the figure is designed to highlight the effect of changing individual ice flow parameters, but that in reality the parameters are not independent.

*Figure 4: Unnecessary; remove.*

We will consider removing figure 4, but note that another group has a paper recently accepted for publication (Igneczi et al 2018) using the transfer functions only along flowlines, and thus feel that mentioning the importance of dimensionality is particularly relevant.

*Figure 5: Unnecessary; remove all except Panel E, which could be incorporated into Figure 3.*

We will remove figure 5 or place it in a supplement, and agree that it is a good idea to place a basal sliding transfer function comparison in figure 3.

*Figure 6: Potentially useful, but why is the misfit pattern so sensitive to $\eta$? How many values of $\eta$ were tested, and why is the misfit so concentrated at 1015 Pa s? Not what I would expect.*

We explored values of effective Newtonian viscosity $\eta$ ranging from $10^1$ to $10^{20}$. Holding other ice flow parameters fixed to reasonable values for our study regions, decreasing $\eta$ generally shifts the transfer function peak to shorter wavelengths. Because our domain sizes (and maximum resolvable wavelengths) are limited, at values of $\eta$ above around $10^{16}$ the transfer peak is shifted to high enough wavelengths that there is essentially no transfer calculated. At viscosities around $10^{15}$ the transfer peak occurs near the longest wavelengths we resolve (around 50-100 km), which does a very poor job of predicting the ice surface. The misfit is particularly large in this part of the parameter space because bed topography amplitude generally increases with wavelength, so a strong expression of these large features creates very unrealistic surface predictions. The changing of the transfer functions becomes much more gradual as effective viscosity decreases beyond around 10^13, which is why there is minimal change in misfit at low values of $\eta$.

*Figure 7: Keep; make color scales the same on Panels E and F.*

Will do.

*Figure 8: Panel B is not useful. Are the data shown in Panel A from this paper, or previous work? Consider deleting entirely.*

The data in both panels are results from our analysis of published data. Panel B is primarily meant to show that some supraglacial stream profiles do exhibit overall concavity (as would be expected in steady state fluvial longitudinal profiles), but that there are many readily visible deviations from this. We do not in detail examine the source of such deviations, except to note that bed topography produces stream profiles with similar properties. We will consider removing panel B, and/or repurposing both panels of the figure to also illustrate the conformity metrics.

*Figure 9: Panel A is misleading because the Stream Free Region (RSF) looks different from all other regions, which may be intended to show the influence of streams. Yet the cause is simply much different H, u, and $\alpha$ in this region compared to other regions (Table 1). For better fidelity, the authors should select a RSF with similar ice geometry to the stream regions.*

This figure is not meant to demonstrate anything about streams, but the point that RSF is too different from our other regions to make such demonstrations is valid. Since it may prove impossible to find replacement regions with ideal attributes and data availability, and since the stream-free region does not contribute significantly to our analysis anyways, we will remove this study region from the revision.

> *Figure 10: Even after a lot of thought, I am still not sure what is being plotted here. I understand the meanings of %d and $\Lambda$, but cannot understand the choice on the y-axis (difference from maximum). In the text, a normalized framework (0 to 1) is discussed, but the data here are not shown that way. The text also highlights variability at small wavelengths (P13 L6-11), but the figure presentation makes this information uninterpretable (all curves are plotted too densely at low wavelengths). Regardless, I infer that the point of this figure is to show the natural variability in both %d and $\Lambda$, by showing the values across R1-R7, and then comparing to the flow networks. For $\Lambda$, the flow networks fall within the natural variability, but for %d, it does not. This could suggest something about the control of fluvial erosion, or other surface processes, on surface topography, but this is not addressed.*

This figure demonstrates that, in both real steam networks and synthetic flow networks, topographic wavelengths between 1-10 km are important and sufficient for explaining stream network structure. We will better describe the metrics and figure in a revision. We normalized the y-axis to highlight that conformity values in all networks plateau and exhibit minimal change as wavelengths smaller than 1 km are added, but will change this if it makes the plots less clear. We will also use larger domains for our synthetic flow networks so that they extend to longer wavelengths to show that topographic wavelengths longer than 19 km are also less significant at controlling stream/synthetic flow network structure. The natural variability in the conformity metrics between regions is interesting, but we do not attempt to explain such variability in this work.

> *Figure 11: Panel A is not necessary, but Panel B presents a comparison of inferred surface to actual surface, which is essential. Why were such comparisons not run on all 7 study regions? Yet, the text (P13 L16-20) declares the slope-area metric to be of limited utility, according to previous work and data from this study. Thus, any conclusions based on this data (P13 L21-27, P14 L1-2) should be de-emphasized or removed.*

We will remove Panel A. We presented synthetic flow network results only in the two study regions where transfer functions produced the best surface predictions, to avoid overly cluttered figures and since we expected these results were most reliable. We will include synthetic flow network results from more regions, perhaps in a supplement. The finding that similar slope-area trends can be produced on surfaces only controlled by bed topography and on fluvially-incised regions of the ice sheet is a key conclusion, since it indicates that if supraglacial fluvial incision has an appreciable impact on stream profiles it is convolved with the effects of variable ice flow. This observation supports our hypothesis that bed topography is the dominant control on stream network structure. Furthermore, it is important to present our observations as a caution that negative slope-area trends on ice sheets do not necessarily imply a landscape shaped by fluvial erosion, as is often considered to be the case in terrestrial landscapes.

> *Figure 12: Keep.*

*I also suggest better distinguishing what is observationally based (e.g., stream networks) from what is computed here using transfer functions (e.g., flow networks).*

We will use a consistent term such as "synthetic supraglacial flow networks" when referring to the flow networks we derive from transfer function predicted surfaces.

*The main conclusion, that "bed topography transfer alone can explain ~1-10 km scale ice sheet surface topography" (P13 L29-30), is not illustrated well in any one figure. It can perhaps be inferred from Figure 7E, but the spatial scale must be eyeballed, rather than shown as the independent variable like in the majority of the figures.*

The point is meant to be illustrated in the spatial domain by figure 7 C,D, and in the spectral domain by figure 9, as well as indirectly by the %downhill and conformity plots. However, we agree that the text is not concisely structured around making this point. We will add panels to figure 9 replicating panels C and D on ice surfaces predicted by the transfer functions to provide a more explicit spectral domain indication that the bed topography can reasonably well explain these wavelengths of surface topography.

*The Discussion section is largely uncoupled from the rest of the work. Speculation on changes in basal slipperiness on hourly to seasonal timescales (P14 L28, L33, P15 L1-2) is not relevant to the bed-to-surface propagation this paper addresses, as the stated timescale for adjustment is >3 years (P14 L14). Thus, the hypothesized feedback (increasing melt changes sliding, which changes IDC size, affects local melt water volumes at the bed, which again changes sliding), which operates on seasonal or shorter timescales, is not supported or constrained by the study. It would be an interesting concept if it could be shown, but that is not accomplished here.*

We feel that section 4.1 of our discussion is justified by our results, and is a natural extension of our results. Section 4.2 makes testable hypotheses if yearly or multi-year averaged ice flow conditions conditions are considered. We will attempt to condense this section to the essential point: "Supraglacial hydrology effects subglacial hydrology (references), which could impact long-term averaged ice flow parameters like basal sliding. If the former connections exist, there could be feedbacks in which subglacial hydrology also effects supraglacial hydrology". We think this point is worth mentioning since it does seem conceivable that there would be some connections between subglacial hydrology and long-term averaged ice flow parameters, and since our work indicates the general effect these ice flow parameters have on supraglacial hydrology. We will consider implementing simple calculations of subglacial hydraulic potential to illustrate how surface topography alone could influence subglacial hydrology.

*The first paragraph of Section 4.1 reads like the main motivation for the study, and as such should appear in the Introduction.*

We will place these statements in the introduction.

*Equations 7 and 8 appear in the Discussion, which is strange, and are not applied to further analysis. They should be removed.*

*The ideas on fluvial erosion (P16 L1-30) are potentially interesting, but again, are completely unexplored in the work. The statement "Our conformity metric calculations (Fig. 10) are consistent with an external control on supraglacial stream network geometry" (P16 L20-21) is*

*not supported by the work. It may or may not be true, but the data were not shown to demonstrate it.*

We will significantly condense and clarify section 4.3. We will place the explanation of the background ideas behind the slope vs drainage area metric in the introduction, which will provide better context as to why we consider this metric in examining how well bed topography can explain supraglacial stream network structure and/or how significant fluvial incision appears to be at shaping such structure. We will clarify the significance of our findings with respect to the slope-area metric, which we discussed earlier in this response, in our results section and/or in this discussion section.

We will also better clarify in the results and/or discussion sections the primary significance of our findings with respect to the conformity metrics, which were also discussed earlier in this response. We will reword "external control" to the intended control of "bed topography transfer". This statement is consistent with our conformity metric results in that the same band of wavelengths can explain stream conformity in real and synthetic supraglacial stream networks, and that these wavelengths correspond well with the wavelengths at which bed topography is predicted to transfer strongly.

*Overall, the base idea is worthy of exploration, but the paper is light on results and heavy on unsupported and speculative discussion, and does not fully consider the limitations of its primary dataset.*

We expect that by restructuring  and being more thorough in our presentation of results we will better emphasise the findings of our work (as discussed above). We will condense and restructure our discussion, so that its contents are more direct extensions of our results. We expect that presenting a condensed form of the argument outlined earlier in the response (and shown in response figure 2) will justify our use of BedMachine mass conservation derived DEMs, and explain that instead producing DEMs with an alternative method such as kriging would not necessarily improve the reliability of our results or significantly affect our conclusions.

References:

Ryser, C., Lüthi, M., Andrews, L., Hoffman, M., Catania, G., Hawley, R., . . . Kristensen, S. (2014). Sustained high basal motion of the Greenland ice sheet revealed by borehole deformation. Journal of Glaciology, 60(222), 647-660. doi:10.3189/2014JoG13J196

MacGregor, J. A., et al. (2016), A synthesis of the basal thermal state of the Greenland Ice Sheet, *J. Geophys. Res. Earth Surf.*, 121, 1328–1350, doi: 10.1002/2015JF003803

Gudmundsson, G. H. (2003), Transmission of basal variability to a glacier surface, *J. Geophys. Res.*, 108, 2253, doi: 10.1029/2002JB002107, B5.

Yang, K., and L. C. Smith (2016), Internally drained catchments dominate supraglacial hydrology of the southwest Greenland Ice Sheet, *J. Geophys. Res. Earth Surf.*, 121, 1891–1910, doi: 10.1002/2016JF003927.

Raymond, M. J., and G. H. Gudmundsson (2005), On the relationship between surface and basal properties on glaciers, ice sheets, and ice streams, *J. Geophys. Res.*, 110, B08411, doi:10.1029/2005JB003681.

Ádám Ignéczi, Andrew J. Sole, Stephen J. Livingstone, Felix S. Ng and Kang Yang. (2018). Greenland Ice Sheet surface topography and drainage structure controlled by the transfer of basal variability. *Front. Earth Sci.* doi: 10.3389/feart.2018.00101

Lindbäck, K., Pettersson, R., Doyle, S. H., Helanow, C., Jansson, P., Kristensen, S. S., Stenseng, L., Forsberg, R., and Hubbard, A. L.: High-resolution ice thickness and bed topography of a land-terminating section of the Greenland Ice Sheet, Earth Syst. Sci. Data, 6, 331-338, https://doi.org/10.5194/essd-6-331-2014, 2014.

Morlighem, M., Rignot, E., Mouginot, J., Seroussi, H., Larour, E. Deeply incised submarine glacial valleys beneath the Greenland ice sheet. *Nature Geoscience.* 2014, 7, 418. http://dx.doi.org/10.1038/ngeo2167

Morlighem, M., E. Rignot, H. Seroussi, E. Larour, H. Ben Dhia, and D. Aubry (2011), A mass conservation approach for mapping glacier ice thickness, *Geophys. Res. Lett.*, 38, L19503, doi:10.1029/2011GL048659.

---

## Author Response (AR1)

**Review Responses and Revision Changes for Manuscript: Basal control of supraglacial meltwater catchments on the Greenland Ice Sheet**

Josh Crozier, Leif Karlstrom, and Kang Yang

We would like to thank both referees for their constructive comments on our submission. We here present point-by-point responses to their comments, and outline the changes we have made in our manuscript accordingly. We expect that these changes have produced a greatly improved manuscript that informs bed-to-surface connections on the Greenland Ice Sheet, and supraglacial hydrology generally. We have attempted to better highlight the contributions of our study (see revised conclusions section).

Key changes in our manuscript include restructuring to better contextualize the geomorphology/meltwater routing parts of our analysis, more thoroughly explaining our model, data, and parameter choices (including addressing seasonality, bed DEM limitations, and slip ratio values), more clearly highlighting our results, and making our figures and discussions more concise.

We note that in our responses all references to specific manuscript sections or figures refer to the revised manuscript unless otherwise indicated, but that text in the Referee comments is unaltered and thus references our original submission .

**Responses to Referee #1:**

*This paper uses previously established transfer functions to make three points:*

1) that 1-10 km scale topography on the ice sheet is controlled by bed topography. I don't disagree with this statement because its more or less the conventional wisdom and others have demonstrated this to be the case. It is certainly not new and I don't feel the results presented really shed any new insight relative to Greenland.

**Response:** That bed topography controls or strongly influences ice surface topography has been mentioned previously in literature. We have cited many of the authors that discuss this link. However, we disagree that our work is not useful in the context of Greenland. We are not aware of any publications explicitly and quantitatively testing this idea using 2D bedrock DEMs, or of any publications quantitatively examining how well bed topography explains 2D ice sheet surface topography as a function of wavelength. Our approach is useful in that we have demonstrated that a relatively simple and easy-to-implement analytical model can explain most IDC-scale (1-10 km) surface topography.

2) Changes in sliding will radically alter the surface topography and catchments, leading to smaller catchments with more moulins and less efficient drainage. This point is somewhat of a stretch given that the high sliding really only occurs in the summer – most of the evolution of the glacier takes place over the other 9 or 10 months of the year.

**Response:** The hypothesis that changes in surface drainage basins could affect subglacial drainage efficiency which would then affect long-term averaged basal sliding is supported by the sensitivity of bed transfer functions to changes in basal sliding. Testing this hypothesis would be an interesting direction for future work. The seasonal dynamics of ice sheets are of course incredibly important, which we acknowledge in sections 2.1 and 2.3.2, but we do not attempt to model such short time scales in this

work. Seasonal dynamics may be part of the reason why the ice surface topography we predict with transfer functions often exhibits not insignificant deviation from surface DEMs.

However, the transfer functions indicate that the wavelengths of features we focus on should change over timescales of 3-60 years, with minimal seasonal variation (see section 2.3.2). We also point out (consistent with the referee's later statement) that we are not aware of observations indicating that IDC-Scale ice surface topography generally changes significantly on a seasonal basis. If the ice surface topography doesn't change significantly on a seasonal basis but other ice flow parameters do vary, then it is reasonable to question how the "long-term effective (or average)" ice flow parameters that govern surface topography relate to the dynamic ice flow parameters (i.e., annual average or peak values). We do not attempt to address this in our work.

Moreover, they appear to use a very high slip ratio of 11 given that from what I can tell its derived using winter velocities in regions with quite warm (perhaps even temperate ice), with high slopes, so one would expect deformation to be significant (~50/50 as Ryser et al, JGlac 2014 show). Ryser et al show slip ratios this high in summer, but only for a few brief peaks each summer (the annual average slip ratio is much lower). Citing this work as well as others on actual slip ratios would make sense. From Figure 6, its seems like the misfit is somewhat insensitive (broad minimum) to this parameter, so how is the ice sheet so sensitive to change in sliding. In short, the feedback they suggest between catchment size and sliding is not at all well supported. It's also not clear how much faith we should put in a theory derived for small perturbations applied to high-amplitude topography with a linear rheology in place of a non-linear rheology. Such cases can be illustrative, but one has to be careful about then inverting and assigning too much quantitative credence to the results.

**Response:** A slip ratio of 11 is indeed at the high end of the ranges presented for our study regions in other publications (such as the suggested Ryser et al 2014 which we have referenced in our revision, or our original reference MacGregor et al 2016), even for summer values. We chose a slip ratio of 11 based on the best fitting values found from inversions in study regions with reliable inversion results. However, the inversion minima is often broad (as indicated by figure 5), and so in some regions we could choose a slip ratio as small as around 4 without obtaining significantly worse surface predictions. We expect that the high values of slip ratio we found primarily reflect the assumption of Newtonian ice rheology that is used in the transfer functions. The analysis of Raymond and Gudmundsson (2005) shows that non-linear rheologies generally increase transfer amplitude peak for a given sliding value, thus linear rheology will predict larger basal slip ratios to attain a given transfer peak amplitude (see section 2.3.3). Raymond and Gudmundsson (2005) also demonstrate that the shape of the transfer function as a function of wavelength is generally quite similar between Newtonian and power law ice rheologies.

Using smaller values for slip ratio would not greatly impact some of our primary conclusions so long at the slip ratios used are not smaller than around 2-4. For the amplitude spectra comparisons we cover in sections 2.3.5, 3.1, and figure 7, using a smaller slip ratio does not strongly change the general shape of the transfer function or location of the transfer amplitude peak. This is implied by figure 3, and we have included a modified version of figure 7 in this response (Response Figure 1) to show that for ice flow parameters representative of our region of interest we still predict similar 1-10 km transfer peaks with a slip ratio of 4. Since the wavelengths of peak bed transfer are relatively insensitive to sliding, our conclusion that bed topography transfer can explain the conformity metrics would still be valid (section 3.2.2 and figure 10). Additionally, the slope of our calculated slope-area trends on synthetic flow networks (figure 9) might decrease if we predict surfaces using lower slip ratios, though the point that bed topography transfer alone can create negative slope-area relations like those observed in stream networks will still be valid (section 3.2.1). Our discussion that focuses in relative changes to basal sliding will also be robust for different baseline sliding choices. As we show in section 4.1 and figure 11, changing slip ratio alters surface topographic basin configuration. This is not inherently inconsistent with the broad constraints that our inversions place on slip ratio, which reflects both bedrock DEM errors and simplifying model assumptions. By showing that different regions of the GIS with a range of ice flow parameters can be reasonably well modeled given sufficiently accurate bed DEMs, we can extrapolate to predict approximate changes in surface topography (and basal hydraulic potential) upon varying ice flow parameters. In other words, we quantify how well the transfer functions work and their sensitivity to bed DEM error, show that they can produce surface topography with sufficient accuracy as to be useful for making general predictions, and then use them as a means to quantitatively predict changes in surface meltwater routing.

**Changes:** In our revision we have used a slip ratio of 10 instead of 11, since 10 is typically considered a more "round" number, and may better imply that we have just picked an approximate value that produces reasonable results in our study regions with the transfer functions we use. We have also attempted to distinctly refer to the "long-term averaged basal sliding parameter" we use in the transfer functions, so as to not imply we expect this parameter is exactly equal to "slip ratio". We have also presented more slip ratio inversion results (table 1). We added discussion of the points listed above in section 2.3.3.

**Response Figure 1.** Demonstration that lower values of slip ratio ( $C^{o^*}$ ) still predict bed topography transfer peaks at wavelengths from 1-10 km (panel B), but less effectively match observed admittances. This figure is similar to figure 10 in our revision except for the additional panel B.

3) There is a lot about thermal-erosion that's not really well explained. There numerous cases where major drainages are observed to be bridged due to large melt channels. So, I am not really sure what the major point is.

**Response:** We are not claiming that large-scale surface topographic features fully control stream network and drainage basin structure, and the possibility that fluvial processes contribute to internally drained basin reorganization on a seasonal scale is not ruled out by our work. Rather, we find that, in all the regions we examined, the general network-wide structure of supraglacial stream networks and the approximate configuration and number density of IDCs can be explained by bedrock transfer. We did not attempt to carry our more rigorous statistical studies over larger regions; this will become more tractable as bed DEM data improves. The fact that we can predict the large-scale basin structure reasonably well from only bed topography, even using a fairly simple ice flow model, verifies that basal processes are the first-order control on IDC-scale surface topography and meltwater routing, a point that has important implications which we explore in our discussion.

**Changes:** We removed extraneous discussions of thermal-erosion, and restructured the rest to better illustrate the relation to bed topography transfer and to our key points. For example, we shifted many points from discussion section 4.3 to section 2.4.1 to better introduce the ideas behind the geomorphological metrics we use.

Nearly every Figure is referenced parenthetically, without ever explaining what the figure is supposed to be showing. Statements like "We computed xyz results to make some point. The results show that. . .." Would be helpful. The captions themselves are generally terse and don't really explain the figures well, especially without supporting explanation in the text. In some cases, the figures appear to be referred to out of order (5 before 4). With respect to the number of figures, this is probably a case of less is more (i.e., fewer, better explained, and more relevant figures).

**Changes:** We better contextualized figure references and added more explanation to the captions. We removed the old figure 2. We simplified figure 4 but left it in place as we believe it provides a useful and concise illustration of the transfer functions that underlie much of our work for unfamiliar readers, and it also shows that for the parameters we are interested in basal sliding variations have a comparatively minimal effect on surface topography. We moved the old figure 4 to a supplement. We modified figure 6 to better illustrate the stream conformity metrics, and removed the tangential stream elevation profile plot. We simplified the transfer results example figure (figure 8). We made the slope-area figure (figure 9) more clear and comprehensive. Lastly, we removed synthetic flow network results from the conformity metrics figure 10) since these did not contribute to the intended point.

The appendix seems to be largely a rehash of Gudmundson's work with a few symbols changed. A whole section to define Fourier transforms is unwarranted.

Changes: We removed these appendices.

*In summary, I don't see that this paper adds much new knowledge or insight in its present form. It probably needs a complete restructuring and rewrite.*

**Response:** We agree that restructuring has make our points more clear and better supported them. Though there is of course much room for future work on the ideas we examine, we do believe that our work makes three primary and worthwhile contributions, as stated in the conclusions (section 5) of our revision.

**Specific Points**

P1/L18 – disperse -> dispersed P1/L18/19 – more dispersed yes, but under the scenarios that would reach this point, the volume of melt water would be greater (i.e., warming world), so it is not clear whether the efficiency would increase or decrease.

**Response:** A good point. While an examination of potential subglacial-supraglacial feedbacks is a natural extension of our work, there are basic aspects of subglacial hydrology that are still poorly known.

**Changes:**We added this to section 4.2 of our discussion. We made clearer in this section that our approach simply indicates that there is plausibly some feedback between surface topography, surface hydrology, and basal hydrology.

P2/L18 – set however off with commas (, however,)

P2 L26/27 – would be appropriate to cite Joughin et al 2013 Cryosphere here (and perhaps elsewhere). Their paper has a quite a bit of discussion on the interaction of basal and surface topography and the effect of water routing.

Changes: Implemented both of the above suggestions.

P2 L31 – insert a comma before "which" P3 L6 – "it is unclear whether dynamic stream incision is efficient enough compared to other topographic influences to influence IDC scale topography and meltwater routing" Not sure I understand this statement – as noted below, a quick google search can turn up many pictures see large stream channels cut by overtopping streams.

**Response:** It has not yet been demonstrated how significant of a role thermal-fluvial incision plays in setting the large-scale structure and evolution of subglacial drainage basins, relative to how much of this structure is primarily set by bed topography. Certainly there are local examples of fluvial erosion influencing drainage patterns, but we show that the effect of bed topography filtered through to the surface appears to be the primary influence on larger scales (longer wavelengths), and is generally of larger amplitude than the effect seasonally averaged fluvial incision has at carving out topography on these scales.

P3 L24 – don't make Greenland Ice Sheet an acronym as GIS is to commonly used for mapping. You are not word constrained and in most cases you can be brief by just saying Greenland or the ice sheet. P2 paragraph that starts with L20 or L26 – there probably should be a reference to Smith, Raymond, and Scambos 2006, JGR F101019 as they look at the transfer of bed topography to the surface of the Greenland ice sheet. Their findings with respect to anisotropy would make sense to discuss later in the paper as well. P3 L31 – replace "resolution" with "posting" as you note in the next couple of sentences the resolution is anything but 150 m. Ditto for P4 L 2, and L3 (using sampling spacing if you want to avoid repetitive use of posting). P4 L4 – "All" to "The" P4 L14 – Define RSF. P5 16 – hyphenate no-flow condition P13 L26 add a "the" before " ~1-10" P13 L30 This is almost identically restates what was said 4 lines earlier. P14 L1-1 again somewhat repetitive and somewhat repeating the obvious that could be inferred from previous work with transfer functions and observations of bed and surface topography.

Changes: Made changes according to the above suggestions.

P14 L16-17 – "If ice surface adjustments to variable basal conditions or ice flow perturbations are sufficiently rapid, surface topographic basin configuration should also vary on seasonal timescales." If this were the case, then such changes should be occurring now. To the extent any such changes have occurred they escaped notice of numerous groups observing elevation time series.

**Response:** We are not aware of any studies that have explicitly examined this (point GPS measurements are not ideal for basin-scale deformation), but if there are generally no significant changes in supraglacial topographic basin configuration then our long-term averaged surface predictions could be considered more robust, since there would be a lower potential for inaccuracy due to seasonal dynamics.

**Changes:** We changed the statement to point out that minimal seasonal adjustment of surface topography is predicted according to the transfer functions.

L14 L24-25 "basal sliding" its important to keep in mind the periods of strong basal sliding relatively brief and most of the year there is no surface melt, so this period of low sliding likely dominates the transfer of bed to surface topography. This statement also applies to the following paragraph. P15-L5-10 – again the winter pattern is likely to dominate and offsets any summer change with a wholesale redistribution of the drainage patterns.

**Response:** We cannot say from our current methods/data if just winter slip ratio influences topography, or if seasonal speedups matter. However, it does not seem unreasonable to propose that summer sliding rates might have some impact on the long-term averaged ice sheet surface topography, unless the ice surface fully readjusts to changing flow each season (which as previously discussed seems unlikely). We additionally note that long-term changes in atmospheric temperatures could change the length of time each year over which increased sliding occurs, whether or not sliding rates are affected.

**Changes:** We clarified in both locations that we are referring to changes to "long term averaged basal sliding".

P15 Section 4.3 There is a significant amount of thermal-fluvial erosion – most stream channels are down-cut by by 10s of centimeters to meters. There are many examples of large stream channels – simple google meltwater stream channels Greenland and select the images tab. The really deep ones are not necessarily that common, but they often occur in locations where a major drainage catchment feeds a lake, that overtops, cut a channel many meters deep, to connect up with another drainage or to find a moulin. I think part of the problem with this section is that its poorly written and its not really clear the point the authors are trying to make.

**Response:** We have indeed published on the potential influence of fluvial erosion on Greenland ice sheet topography (Karlstrom et al., JGR, 2013, Karlstrom and Yang, GRL, 2016). This influence is undeniable on small scales. But bed topography filtered through to the surface is of larger amplitude than seasonal fluvial-incision in most places, so basin-scale structures are essentially static year to year - this may also be seen clearly on Google Earth (also see Fig 1 of Karlstrom and Yang 2016). We are focusing on what primarily controls basin-scale meltwater routing here, and restructured the text to make this more clear.

**Responses to Referee #2:**

**Summary**

This paper explores the factors controlling the catchments of surface rivers on the western Greenland Ice Sheet. It focuses on the relationship between basal topography and these rivers, and concludes that certain geometric aspects (basal bumps and the basal slip ratio) control the organization of surface hydrology. From there, a possible ice-flow feedback is hypothesized based on future projected changes in melt rate and slip ratio. The sign of the ice-flow feedback is unknown.

The study emphasizes the methods (Laplace-domain transfer functions) and most of the results presented (transfer function amplitudes) are a step away from reality, limiting the extent to which results are compared to data. Accordingly, the Results section is very brief (2 pages) compared to the rest of the manuscript (17 pages + Appendix) and the Methods section (8 pages). Phrases like "as expected" of "consistent with previous work" appear frequently, highlighting that this study is light on novel contributions. Most of the 3-page Discussion is speculative and only loosely constrained by the results presented.

**Response:** We attempted to use the best data-sets available, but given that there were still limitations in data quality for important factors such as bed elevation, we chose modeling approaches (basal transfer functions and surface flow routing) that are simple to interpret, apply, and generalize despite limitations in accuracy. Our evaluation and presentation metrics (amplitude spectra, slope-vs-drainage area trends, and stream network conformity values) capture general traits of surface topography and stream networks that are robust in data and should not depend greatly on our model simplifications. These metics form the basis for a quantitative verification, using new datasets over multiple regions of the Greenland Ice Sheet ablation zone, of the extent to which bed topography explains surface topography and meltwater routing. Such a verification sets the stage for future more fully mechanistic studies. Our approach also permits the (testable) prediction of surface topography and drainage basin configurations in different ice flow conditions. These predictions indicate that changing ice flow conditions can appreciably affect supraglacial IDC configurations, which is a novel and significant point that we hope will spark further study.

**Changes:** Our revision includes a more thorough presentation of results and a restructured discussion with explicit calculations of subglacial hydraulic flow pathways and supraglacial IDC configurations (sections 4.1 and 4.2).

The study design is flawed in that the root data (Morlighem bed DEM) are not independent of the validation data (ArcticDEM for the ice-sheet surface) in the regions the authors chose to study (which are, incidentally, areas where Morlighem applied mass conservation). The authors also studied one area (R7) where mass conservation was not applied; results there are not shown, but I would expect their predicted surface to more poorly match the true (Arctic DEM) surface. This is hinted at in Figure 9, but never addressed. The techniques used to generate the bed DEM must be considered in this analysis; preferably, multiple regions with bed DEM constructed from mass conservation and with kriging should be analyzed and compared to one another.

**Response:** The primary regions we examine are areas where mass conservation (based on surface elevation, velocity, and surface+bed mass balances) is used in conjunction with radar data to derive BedMachine DEMs. As discussed in section 2.2, we are interested in regions that exhibit significant supraglacial stream network development (typically at moderate elevations), near-uniform surface velocities, and that have high resolution surface DEMs. Such regions of the ice sheet seem to generally be where mass conservation instead of kriging was used. However, we expect that BedMachine is the best choice for our study due to three reasons:

(1) As far as we are aware, BedMachine is the most accurate Greenland bed DEM currently available, due in part to its use of mass conservation modeling which has advantages described in Morlighem et al (2011) and Morlighem et al (2014).

(2) The method used in derivation of the BedMachine DEM only considers mass conservation, and is thus fundamentally different from the Gudmundsson transfer functions which are derived from both mass conservation and the Stokes flow equations.

We provide a thought experiment to demonstrate this distinction: given a bed elevation DEM, we could apply the BedMachine mass conservation method in reverse to predict steady-state surface elevations. Doing this uniquely would require a full surface velocity map, in addition to target values of background thickness and surface slope (and basal/surface mass balances if those are not assumed to be zero). The transfer functions make surface elevation predictions given the same target thickness and slope values, but just a single background surface velocity vector and a value/values for slip ratio. The transfer functions are thus also independently solving for a full velocity field by incorporating approximations of how ice should flow in response to gravity and pressure gradients. If we fed the same data we use to implement the transfer functions into an "inverse BedMachine model", by using single velocity vector and uniform mass balances over the whole domain, the methods would in general not predict the same surfaces.

By verifying that the transfer functions can reasonably well predict the ice surfaces in our study regions, what we are verifying is that the approximations used to derive of the transfer functions are reasonably effective at least over 1-10 km scales. Furthermore, since where mass conservation modeling was used in the BedMachine DEM the mass balance terms were not perturbed to account for spatially nonuniform processes like fluvial incision, we are verifying that most of the ice surface topography at these scales is consistent with ice flow alone.

(3) We included regions with relatively dense radar coverage, so the influence on mass conservation derived DEMs of surface data will be limited. The analysis we use for testing the possible effect of bed DEM error on our surface predictions, covered in section 2.3.4, 3.1 (see figures 2 and 8.D) provides an indirect indicator of how sensitive our surface predictions are to the mass conservation modeling used in the BedMachine DEMs. This is the case because the published BedMachine error generally increases with increasing distance from radar data points. See the included response figure 2 for further elaboration on this. Our error analysis thus essentially demonstrates that in regions of dense radar coverage, the large-scale surface depressions/ridges we focus on should not be too significantly influenced by the topography between radar transects, and thus by the extrapolation method is used.

We could attempt to obtain raw radar data for our study areas and interpolate it into DEMs using a method like kriging, but this would mean using less accurate bed elevation values (Morlighem et al 2014), and it is not clear that doing so would significantly impact our results or interpretations.

---

## Referee Report (RR1)

**Review for "Basal control of supraglacial meltwater catchments on the Greenland Ice Sheet" by Josh Crozier, Leif Karlstrom, and Kang Yang**

**General Comment**

This is an interesting paper that presents original research on how the topography of the bed below the Greenland ice sheet shapes the overlying surface ice topography, and how this in turn controls supraglacial internally drained configurations (IDC) as well as subglacial drainage patterns. The scope of the paper is broad, and the research is detailed and well executed. The authors combine a number of techniques, including remote sensing DEMs and radar, upward continuation of bed topography, and fluvial morphometric analysis, to interrogate the topographic controls on IDC and how they depend on ice flow characteristics, as well as their role in establishing subglacial meltwater pathways. There is a lack of studies that address the broad aspects of glacial hydrology, in particular the interplay between supraglacial and subglacial drainage and their connection to bed and surface ice topography, so this is a welcome contribution. The paper is well written and structured. I do have two general comments and some minor issues I would like to see clarified prior to publication:

- Supraglacial flow direction: Throughout the text, there appears to be several contradictions stating the direction of supraglacial drainage (i.e., line 33 "stream channels () flow in directions not parallel to the surrounding ice surface slope or slice through topographic ridges", but then lines 26-29 in page 12 "%d should generally increase and approach 100% since streams do not flow uphill. Similarly, $\Delta$ should generally increase and approach 1, since water should generally flow in the direction of steepest descent." Please clarify throughout the text.

- Figures: Figures should be larger, it is hard to see some of the details. Font sizes should be the same for the different labels, and label quality seems compromised in Fig. 2 and Fig. 9. See my specific comments below for other small issues.

**Specific comments**

Page 1 line 7: "a suite of recent datasets" reads vague, rephrase to make it more specific

Page 2 line 13: Can you give numbers for gradual and steep surface slope?

Page 3 line 4: Reference missing after "meltwater routing"

Page 3 line 26-27: Add a few words at the end of the introduction about examples of IDC and their spatial scales

Page 4 line 10: "Bed DEM". Correct typo.

Page 10 line 7: The stream power law was originally proposed by [1].

Page 10 line 20: Values of n for equation 7 are given, but the choice of the power m is not discussed in the paper. Can you add one or two lines in this paragraph about your choice of m, and how it may influence your results? In river systems, m depends on both hydraulic geometry (which is empirically different in supraglacial systems, see [2, 5] and an empirical basin hydrology relationship [4, 3]. I am curious about how to estimate m in supraglacial channels

Page 13 line 29: Is the assumption of essentially zero effective basal pressure consistent with your choice of C0? Please discuss in one or two lines.

Page 15 line 12: The discussion of the slope-area exponent is distracting, I would suggest removing or reducing it.

Page 15 line 24: Fix typo "slope-are".

Page 15 lines 28-33: This paragraph got me a bit confused  see marked-up pdf attached. Throughout the text, you are using fluvial erosion models (stream power law) based on the fact that supraglacial channels behave fluvially. The sentence "slope-area discrepancies that could indicate a fluvial signature in observed stream networks" seems to contradict this fluvial model adopted throughout the paper. Please clarify or rephrase the writing.

Page 16 line 10: Give more details about the "changes" you are referring to  the sentence reads vague.

Page 16 line 16: It should be "affected" and not "effected". Please fix typo.

Page 16 line 29: fix typo: "IDC-scale scale".

Page 17 line 9: Add numbers for the basin density, either here in the text or in Fig. 11. This will help the comparison among basins, which is already done in qualitative terms.

Page 17 line 11: Missing "that" in "expect changes in topographic basin density predicted with changing ice flow conditions should generally correspond".

Page 17 lines 12-14: In this discussion, it would help the reader if you refer directly to the panels in Fig. 11.

Page 18 line 11: The wording of this sentence suggests that a rapid development of subglacial channels increases subglacial pressure. I suggest changing 'rapid' to 'slower'.

Page 18 line 17: Fix typo: "effecting" should be "affecting".

Page 18 line 24: I believe you mean Fig. 10 instead of Fig. 11

**Figures and tables:**

Fig. 1: Can you make the white arrows a bit larger, or increase the white contrast? It appears that both figures (in particular B) have been shrunk horizontally. Could you fix it to make panel B a bit wider?

Fig. 2: Give references for the data sources, even if not directly mentioned, if they are used in the study.

Fig. 3: This figure would benefit if enlarged. Consider having two panels in vertical by two in horizontal.

Fig. 4: Color scale labels should be the same font size than x and y axis labels, and bigger panels would make visualization better.

Fig. 5: Font sizes should be the same for the x, y, labels and the color bar.

Fig. 6: Consider adding the 6 km cutoff filter on the title of panel B.

Fig. 7: What is the shaded part in the plot? Consider either removing it or explaining its significance in the caption, as appropriate. Also missing a label for the x axis.

Fig. 8: Consider changing the title of panel C to "misfit", and detail the misfit calculation (as it is now in the title of panel C) in the caption. Consider, additionally, showing the percentage misfit in this panel, as this is the metric discussed in the results.

Fig. 9: Larger overall figure and larger labels would help to visualize the results.

Fig. 10: Both panels have shaded areas on them please explain what they mean in the caption or remove them, as appropriate.

Fig. 11: The discussion of the results of this figure is about basin densities, but no numbers are given as a reference. I would suggest to give the basin density measurements for each scenario, and give them in the caption.

Table 1: Lines 2 and 3 seem to belong to the main text, not the table caption. Consider relocating them.

**Anna Grau Galofre**

**References**

[1] A. D. Howard and G. Kerby. Channel changes in badlands. *Geological Society of America Bulletin*, 94 (6):739–752, 1983.

[2] R. A. Marston. Supraglacial stream dynamics on the Juneau Icefield. *Annals of the Association of American Geographers*, 73(4):597–608, 1983.

[3] G. Parker, P. R. Wilcock, C. Paola, W. E. Dietrich, and J. Pitlick. Physical basis for quasi-universal relations describing bankfull hydraulic geometry of single-thread gravel bed rivers. *Journal of Geophysical Research: Earth Surface*, 112(F4), 2007.

[4] K. X. Whipple and G. E. Tucker. Dynamics of the stream-power river incision model: Implications for height limits of mountain ranges, landscape response timescales, and research needs. *Journal of Geophysical Research: Solid Earth*, 104(B8):17661–17674, 1999.

[5] K. Yang, L. C. Smith, V. W. Chu, L. H. Pitcher, C. J. Gleason, A. K. Rennermalm, and M. Li. Fluvial morphometry of supraglacial river networks on the southwest Greenland Ice Sheet. *GIScience & Remote Sensing*, 53(4):459–482, 2016.

---

## Author Response (AR2)

**2nd Review Responses and Revision Changes for Manuscript: Basal control of supraglacial meltwater catchments on the Greenland Ice Sheet**

Josh Crozier, Leif Karlstrom, and Kang Yang

We thank both reviewers for their helpful suggestions, and expect that implementing the minor changes suggested has resulted in a well-polished manuscript.

**Response to Report #1:**

*This is an interesting paper that presents original research on how the topography of the bed below the Greenland ice sheet shapes the overlying surface ice topography, and how this in turn controls supraglacial internally drained configurations (IDC) as well as subglacial drainage patterns. The scope of the paper is broad, and the research is detailed and well executed. The authors combine a number of techniques, including remote sensing DEMs and radar, upward continuation of bed topography, and fluvial morphometric analysis, to interrogate the topographic controls on IDC and how they depend on ice flow characteristics, as well as their role in establishing subglacial meltwater pathways. There is a lack of studies that address the broad aspects of glacial hydrology, in particular the interplay between supraglacial and subglacial drainage and their connection to bed and surface ice topography, so this is a welcome contribution. The paper is well written and structured. I do have two general comments and some minor issues I would like to see clarified prior to publication:*

*Supraglacial flow direction: Throughout the text, there appears to be several contradictions stating the direction of supraglacial drainage (i.e., line 33 "stream channels () flow in directions not parallel to the surrounding ice surface slope or slice through topographic ridges", but then lines 26-29 in page 12 "%d should generally increase and approach 100% since streams do not flow uphill. Similarly, conformity factor should generally increase and approach 1, since water should generally flow in the direction of steepest descent." Please clarify throughout the text.*

**Response**: From mapview supraglacial streams may appear to flow "uphill" (or in directions not aligned with the direction of steepest descent) with respect to surrounding large scale topography, such as in the cases we mention where streams incise narrow channels through larger topographic ridges. However, given high enough resolution DEMs to resolve the actual stream channels in such cases (for which the 2m DEMs we use are generally sufficient), we would expect that the actual thalwegs (lines following the deepest part of stream channels) are not routing uphill, since water itself should not flow uphill (except at the very local scale of channel features such as bedforms). Our conformity metrics are designed precisely to quantify how prevalent such deviations between stream-channel pathways and regional topography are at a given topographic filter wavelength.

**Changes**: We changed wording in several places to clarify this.

> *Figures: Figures should be larger, it is hard to see some of the details. Font sizes should be the same for the different labels, and label quality seems compromised in Fig. 2 and Fig. 9. See my specific comments below for other small issues.*

**Changes**: We made font sizes the same for all labels on each figure, enlarged several of the figures, and uploaded higher quality versions of figures 2 and 9.

> *Specific comments*
> *Page 1 line 7: "a suite of recent datasets" reads vague, rephrase to make it more specific*

**Changes**: Fixed

> *Page 2 line 13: Can you give numbers for gradual and steep surface slope?*

**Changes**: Done

> *Page 3 line 4: Reference missing after "meltwater routing"*

**Changes**: Added a relevant reference

> *Page 3 line 26-27: Add a few words at the end of the introduction about examples of IDC and their spatial scales*

**Changes**: Added spatial scales

> *Page 4 line 10: "Bed DEM". Correct typo.*

**Changes**: Fixed

> *Page 10 line 7: The stream power law was originally proposed by [1].*

**Changes**: Added this reference

> *Page 10 line 20: Values of n for equation 7 are given, but the choice of the power m is not discussed in the paper. Can you add one or two lines in this paragraph about your choice of m, and how it may influence your results? In river systems, m depends on both hydraulic geometry (which is empirically different in supraglacial systems, see [2, 5] and an empirical basin hydrology relationship [4, 3]. I am curious about how to estimate m in supraglacial channels*

**Response**: We do not assume or use a particular value of *m* in this work, but just introduce the stream-power equation to to motivate our examination of stream elevation profiles and slope-area relationships. We would point to the the supplemental information of Karlstrom and Yang (2016) for an explanation/equations of the factors controlling *m* in supraglacial streams. If it is assumed that supraglacial stream elevation profiles are incisionally controlled, that steam profiles are in a steady-state (at least relative to whole surface lowering), and that ice flow produces something analogous to uniform uplift, then the power-law fit exponents we show in figure 9 would provide an estimate for *-m/n*. Because n=1 for stream power in a thermally eroding channel [Karlstrom and Yang 2016], the fits in figure 9 might be interpreted as reflecting m directly. However, we do not expect that these assumptions are generally accurate in the Greenland ice sheet ablation zone because stream profile slope-area relations also reflect contributions from bed topography and ice flow. We now state this in the text.

**Changes**: We added brief mention of previously estimated values of *m* and of the factors m depends upon.

> *Page 13 line 29: Is the assumption of essentially zero effective basal pressure consistent with your choice of C0? Please discuss in one or two lines.*

**Changes**: Added lines indicating that this probably isn't consistent with our choice of C0, but that we chose zero effective pressure to examine the maximum feasible impacts on basal hydraulic potential from changing surface topography.

> *Page 15 line 12: The discussion of the slope-area exponent is distracting, I would suggest removing or reducing it.*

**Changes** (assuming Page 15 line 25 was the intended reference): We made the reference to slope-area exponents more explicit, and moved it to the figure caption as this information is useful for interpreting the figure.

> *Page 15 line 24: Fix typo "slope-are".*

**Changes**: Fixed

> *Page 15 lines 28-33: This paragraph got me a bit confused see marked-up pdf attached. Throughout the text, you are using fluvial erosion models (stream power law) based on the fact that supraglacial channels behave fluvially. The sentence "slope-area discrepancies that could indicate a fluvial signature in observed stream networks" seems to contradict this*
> *uvial model adopted throughout the paper. Please clarify or rephrase the writing.*

**Changes**: Changed wording to indicate that by discrepancies we meant differences between the slope area relations predicted without any fluvial incision (on bed topography transfer predicted ice surfaces) and those observed in supraglacial stream networks.

> *Page 16 line 10: Give more details about the "changes" you are referring to the sentence reads vague.*

**Changes**: Fixed

> *Page 16 line 16: It should be "affected" and not "effected". Please fix typo.*

**Changes**: Fixed

> *Page 16 line 29: fix typo: "IDC-scale scale".*

**Changes**: Fixed

> *Page 17 line 9: Add numbers for the basin density, either here in the text or in Fig. 11. This will help the comparison among basins, which is already done in qualitative terms.*

**Changes**: We added basin density values in the text.

> *Page 17 line 11: Missing "that" in "expect changes in topographic basin density predicted with changing ice flow conditions should generally correspond".*

**Changes**: Fixed

> *Page 17 lines 12-14: In this discussion, it would help the reader if you refer directly to the panels in Fig. 11.*

**Changes**: Done

> *Page 18 line 11: The wording of this sentence suggests that a rapid development of subglacial channels increases subglacial pressure. I suggest changing 'rapid' to 'slower'.*

**Changes**: Fixed

> *Page 18 line 17: Fix typo: "effecting" should be "affecting".*

**Changes**: Fixed

> *Page 18 line 24: I believe you mean Fig. 10 instead of Fig. 11*

**Response**: Figure 11 is the intended reference, as this is the figure showing the affects various ice flow parameters have on surface topographic basin density

> *Figures and tables:*
> *Fig. 1: Can you make the white arrows a bit larger, or increase the white contrast? It appears that both figures (in particular B) have been shrunk horizontally. Could you fix it to make panel B a bit wider?*

**Changes**: Fixed

> *Fig. 2: Give references for the data sources, even if not directly mentioned, if they are used in the study.*

**Response**: We referenced all data plotted in this figure, the BedMachine DEM and the Cresis radar picks.

> *Fig. 3: This figure would benefit if enlarged. Consider having two panels in vertical by two in horizontal.*

**Changes**: Enlarged figure

> *Fig. 4: Color scale labels should be the same font size than x and y axis labels, and bigger panels would make visualization better.*

**Changes**: Fixed

> *Fig. 5: Font sizes should be the same for the x, y, labels and the color bar.*

**Changes**: Fixed

> *Fig. 6: Consider adding the 6 km cutoff filter on the title of panel B.*

**Changes**: Fixed

> *Fig. 7: What is the shaded part in the plot? Consider either removing it or explaining its significance in the caption, as appropriate. Also missing a label for the x axis.*

**Changes**: Fixed

> *Fig. 8: Consider changing the title of panel C to "misfit", and detail the misfitt calculation (as it is now in the title of panel C) in the caption. Consider, additionally, showing the percentage misfit in this panel, as this is the metric discussed in the results.*

**Changes**: We re-titled panel C, but left the misfit unscaled as this allows for a more direct comparison of the misfit map with the two surfaces. We described misfit in the caption.

> *Fig. 9: Larger overall figure and larger labels would help to visualize the results.*

**Changes**: Fixed

> *Fig. 10: Both panels have shaded areas on them please explain what they mean in the caption or remove them, as appropriate.*

**Changes**: Fixed

> *Fig. 11: The discussion of the results of this figure is about basin densities, but no numbers are given as a reference. I would suggest to give the basin density measurements for each scenario, and give them in the caption.*

**Changes**: We added basin density values to the main text, but left the counts in the figure titles since the counts are these are more directly visible in the plots, and since including both counts and densities in the titles seems redundant.

> *Table 1: Lines 2 and 3 seem to belong to the main text, not the table caption. Consider relocating them.*

**Changes**: We consolidated these lines in the caption, and mentioned in the text that our misfit metric is not necessarily a comprehensive indicator of fit quality.

**Response to Report #2:**

> *This manuscript is remarkably improved and approaching publication readiness, in my opinion. Overhaul of the figures has greatly improved the clarity of the paper, and the reorganizing and re-positioning of text has resulted in a more balanced manuscript in terms of methods, results, and discussions. The addition of new results with respect to fluvial incision has also provided a useful complement to the basal transfer component.*

> *I am lukewarm on the results and conclusions regarding IDC density (Figure 11). Future changes in IDC density become one of the primary conclusions of the work, yet the IDC number density is not robust to changes in the parameter space explored here. Further explanation, at least, is needed (see specific comment below).*

**Response**: This was a plotting error, the surface prediction under current ice flow conditions (panel B) was actually made with a lower slip ratio. The other panels were correct, and with the correct prediction in panel B it is apparent that there are relatively consistent trends in surface basin density with changing ice flow parameters. We carried out this parameter study in our

study region R2 as well, and find similar trends in basin density with each ice flow parameter, indicating that at least the general direction and order of magnitude of such trends is robust.

**Changes**: We fixed the error so that the correct prediction is now used in panel B.

*Figure 11 It is strange that any change (increase / decrease) in any of the 4 parameters causes an increase in the number of topographic basins, and makes me question the robustness of this particular analysis. Why is it that the base-case parameters chosen occupy a local minimum in IDC number density? This number density does agree with observations (Figure 11a), but the controls on IDC density are not well illuminated by this set of results.*

**Response**: see response and changes for above comment.

*A short list of particular comments follows. I am confident the authors can address these remaining critiques without many additional substantial changes to the work and presentation.*

*P8 L13-21 This paragraph makes the role of C\* and the authors' decision path on C0\* versus C\* unclear. The use of 3 different terms, C0, C\*, and C0\*, if ultimately the chosen slip ratio is spatially constant, could probably be simplified.*

**Changes**: We have tried to make more clear in several places that $C^{0*}$ refers to a uniform background sliding parameter value, and that $C^*(x,y)$ refers to a spatially variable sliding parameter. In this paragraph we clarify that we are showing that the effects of spatial variability in $C^*(x,y)$ are negligible in our study regions, and that we thus just assume uniform $C^*(x,y) = C^{0*}$ elsewhere in our analysis.

*P9 L24-33 The descriptions in this paragraph (description of Figure 7) would benefit from being better anchored to separate the methods used for Figure 7b from those used for Figure 7c. Could the 1D and 2D DFTs be given separate paragraphs in 2.3.2, more descriptive and separate captions in Figure 7c, and some emphasis in this paragraph that they are independent of the transfer function result (Figure 7a) to which they are being compared.*

**Changes**: We have emphasized that the observed/empirical admittance we are calculating is distinct from the predicted transfer amplitude given by the analytical transfer functions. We made separate paragraphs (and separate lines in the figure caption) for each observed/empirical admittance calculation method, and described both methods more explicitly.

*P14 L11-19 It would benefit to step through each panel of Figure 7 here.*

**Changes**: We made more explicit interpretations and comparisons between the figure panels.

> *Figure 7 - x axes are missing label and scale.*

**Changes**: Fixed

> *Figure 8 / P14 L28 What is the mean misfit in R1?*

**Changes**: We added text to the caption indicating mean misfit is 13.3% of regional topographic relief.

> *P14 L30 "unaccounted for physics" is a poor catch-all; what do you mean apart from the linear rheology that you already described? Revise or remove*

**Changes**: We changed wording to "unaccounted for processes such as fluvial incision and propagating kinematic ice waves"

> *P17 L4 Some extra words in here*

**Changes**: Fixed

> *P19 L2-3 Rephrase this sentence to make it clear that bed topography is the dominant influence at this scale, and fluvial erosion is secondary.*

**Changes**: Fixed

> *P19 L11-12 Suggest adding "...IDC scales, as our results suggest, then supraglacial stream incision..."*

**Changes**: Fixed

**References**

[revised manuscript text omitted]
$ (m) | $U$ (m/yr) | $\alpha$ (rad) | RACMO melt rate (mm water/yr) | Bed DEM mean error (m) | Surface prediction mean misfit ($C^{0*} = 10$) (% topographic relief) | Best fitting $C^{0*}$ |
|---|---|---|---|---|---|---|---|
| R1 | 824 | 86 | 0.013 | 2107 | 28.9 | 13.3 | 10 |
| R2 | 1075 | 87 | 0.009 | 1145 | 30.0 | 9.35 | 6 |
| R3 | 1233 | 88 | 0.009 | 793 | 41.1 | 10.4 | 13 |
| R4 | 1335 | 81 | 0.006 | 556 | 80.0 | 12.4 | 29 |
| R5 | 840 | 214 | 0.017 | 1321 | 55.8 | 13.9 | 23 |
| R6 | 1266 | 87 | 0.007 | 711 | 53.2 | 11.2 | 11 |
| R7 | 1187 | 92 | 0.011 | 1552 | 49.1 | 13.7 | 11 |